# PARG is essential for Polθ-mediated DNA end-joining by removing repressive poly-ADP-ribose marks

Umeshkumar Vekariya[1,6], Leonid Minakhin[2,6], Gurushankar Chandramouly[2], Mrityunjay Tyagi[2], Tatiana Kent[2], Katherine Sullivan-Reed[1], Jessica Atkins[1], Douglas Ralph[2], Margaret Nieborowska-Skorska[1], Anna-Mariya Kukuyan[1], Hsin-Yao Tang [3], Richard T. Pomerantz [2] ✉ & Tomasz Skorski [1,4,5] ✉

DNA polymerase theta (Polθ)-mediated end-joining (TMEJ) repairs DNA double-strand breaks and confers resistance to genotoxic agents. How Polθ is regulated at the molecular level to exert TMEJ remains poorly characterized. We find that Polθ interacts with and is PARylated by PARP1 in a HPF1-independent manner. PARP1 recruits Polθ to the vicinity of DNA damage via PARylation dependent liquid demixing, however, PARylated Polθ cannot perform TMEJ due to its inability to bind DNA. PARG-mediated de-PARylation of Polθ reactivates its DNA binding and end-joining activities. Consistent with this, PARG is essential for TMEJ and the temporal recruitment of PARG to DNA damage corresponds with TMEJ activation and dissipation of PARP1 and PAR. In conclusion, we show a two-step spatiotemporal mechanism of TMEJ regulation. First, PARP1 PARylates Polθ and facilitates its recruitment to DNA damage sites in an inactivated state. PARG subsequently activates TMEJ by removing repressive PAR marks on Polθ.

DNA polymerase theta (Polθ, encoded by *POLQ* gene) is a unique DNA helicase-DNA polymerase fusion protein that promotes error-prone repair of DNA double-strand breaks (DSBs) by a mechanism referred to as Polθ-mediated end-joining (TMEJ), or microhomology-mediated end-joining (MMEJ) and alternative end-joining (Alt-EJ)[1–3]. Polθ also performs translesion synthesis and recent studies indicate it additionally performs DNA repair synthesis at single-strand DNA (ssDNA) gaps[4–9]. Polθ-mediated DNA repair activities have been reported in S, G2 and M cell cycle phases[4,10–14].

Overexpression of Polθ in cancers is associated with a poor prognosis[15–17], and Polθ confers resistance to genotoxic cancer therapeutics such as ionizing radiation, bleomycin and topoisomerase inhibitors[3,18,19]. In addition, Polθ protects cancer cells from DNA damage caused by metabolically generated toxic aldehydes[20]. Inhibition of Polθ is synthetic lethal with homologous recombination (HR) factors BRCA1/2 and other DNA damage response (DDR) proteins[2,10,21]. Thus, Polθ is regarded as an important cancer drug target, and three Polθ inhibitors are in clinical trials (NCT04991480, NCT05898399, NCT06077877).

The TMEJ pathway acts on 3′ ssDNA overhangs at DSBs which are promoted by the MRE11-RAD50-NBS1-CtIP nuclease complex during S/G2 cell cycle phases[17]. The A-family polymerase domain of Polθ binds 3′ ssDNA overhangs and facilitates synapsis between 3′ ssDNA overhangs possessing minimal homology between bases, referred to as microhomology[1,22]. The enzyme then extends the minimally paired ssDNA overhangs, resulting in stabilization of the DNA synapsis. The

[1]Fels Cancer Institute for Personalized Medicine, Lewis Katz School of Medicine, Temple University, Philadelphia, PA 19140, USA. [2]Thomas Jefferson University, Sidney Kimmel Cancer Center, Department of Biochemistry and Molecular Biology, Philadelphia, PA 19107, USA. [3]Proteomics and Metabolomics Facility, The Wistar Institute, Philadelphia, PA 19104, USA. [4]Department of Cancer and Cellular Biology, Lewis Katz School of Medicine, Temple University, Philadelphia, PA 19140, USA. [5]Nuclear Dynamics and Cancer Program, Fox Chase Cancer Center, Philadelphia, PA, USA. [6]These authors contributed equally: Umeshkumar Vekariya, Leonid Minakhin. ✉e-mail: richard.pomerantz@jefferson.edu; tskorski@temple.edu

superfamily 2 helicase domain of Polθ (Polθ-hel) binds various DNA substrates, exhibits ATPase activity that is stimulated by ssDNA, promotes ATP-independent ssDNA annealing, and displays relatively weak ATP-dependent 3′–5′ DNA unwinding activity[23–25]. Despite the discovery of these biochemical activities of Polθ-hel, its function in TMEJ remains unclear. For example, biochemical studies found that the ATPase function of Polθ does not contribute to the enzyme's end-joining activity[1]. However, studies in mammalian cells indicate that Polθ-hel promotes TMEJ via ATP-dependent dissociation of RPA from 3′ ssDNA overhangs[25]. Moreover, Polθ-hel has been implicated in dissociating RAD51-ssDNA filaments[21]. Additional studies in *Drosophila* found that Polθ's ATPase activity contributes to the repair of interstrand crosslinks, but not radiation-induced DSBs[26].

Although the biochemical and cellular activities of Polθ have been widely studied, little is known about how Polθ is regulated at the molecular level. Prior studies demonstrated that accumulation of Polθ at laser and ultraviolet light (UV)-induced DNA damage was independent of ATM and ATR signaling, but was significantly reduced by inhibition or suppression of Poly(ADP-ribose) polymerase 1 (PARP1)[2,21,27,28]. PARP1 suppression has also been shown to inhibit TMEJ in mammalian cells which supports a regulatory role for PARP1 in recruiting Polθ to DNA damage[21,28]. Whether PARP1 acts on upstream factors to facilitate Polθ recruitment to DNA damage or regulates Polθ through direct interactions is not known. Lastly, separate studies claim opposing (stimulatory and inhibitory) roles for PARP1 in the 5′–3′ DNA resection process which is required for TMEJ[27,29].

PARP1 dependent poly-(ADP)-ribosylation (PARylation) of proteins is one of the major post-translational modification events involved in the DNA damage response (DDR)[30,31]. PARP1 and poly(ADP-ribose) glycohydrolase (PARG) are the key enzymes orchestrating the poly(ADP)-ribosylation kinetics of DDR proteins. For example, PARP1—the founding member of the poly(ADP-ribose) polymerase (PARP) family—transfers adenosine diphosphate (ADP)-ribose from nicotinamide adenine dinucleotide (NAD$^+$) to substrate proteins enabling their PARylation following DNA damage which contributes to relaxation of local chromatin and recruitment of chromatin modulators and DDR proteins to DNA damage foci[32].

PARP1 has been studied mostly in the context of PARylation on aspartate and glutamate[33]. However, after DNA damage, PARylation occurs predominantly on serine residues[34]. PARP1 requires an accessory factor HPF1 for efficient addition of single units of ADP-ribose to serine residues following by its rapid dissociation, allowing PARP1 to extend initial reaction to poly-ADP-ribose[35,36].

PARylation is tightly controlled by the glycohydrolase activity of PARG which is also recruited to DNA damage sites[37,38]. Thus, the interplay between PARP1 and PARG is thought to regulate the PARylation status of relevant DDR proteins[39], and multiple studies demonstrated that both PARP1 and PARG contribute to DDR[38,40,41]. Although PARP1 is known to facilitate protein recruitment to DNA damage sites via PARylation-dependent liquid-liquid phase separation (LLPS)[42], how PARG dePARylation activity contributes to DDR at the molecular level remains unclear.

However, while PARG is critical for cleaving the bond between poly-ADP-ribosylation subunits, it is unable to remove mono-(ADP)-ribose from a protein[43]. Instead, mono-ADP-ribosylated (MARylated) serines are deMARylated by ADP-ribosylhydrolase 3 (ARH3)[44]. While persistent PARylation is toxic, endogenous MARylation persists on chromatin throughout the cell cycle and is well tolerated[45].

Here, we investigated whether the interplay of PARP1 and PARG is important for the regulation of TMEJ and the specific activities of Polθ. We find that PARP1 binds to and directly PARylates Polθ in vitro and in cells. However, PARylated Polθ (PAR-Polθ) is unable to perform TMEJ in vitro due to its inability to bind DNA despite its PARylation-dependent recruitment to DNA-enriched liquid condensates by PARP1. Hence, PARG is needed to reactivate Polθ DNA binding and its TMEJ

activity by removing repressive PAR marks on Polθ. In support of this, cellular studies show that PARG is essential for TMEJ and supports a two-step mechanism of a PARP1-PARG regulatory axis of Polθ and TMEJ. In the first step, the rate of PARP1-Polθ spatiotemporal recruitment to DNA damage foci supports a rapid step whereby PARP1 PARylates Polθ and facilitates its recruitment to the vicinity of DNA damage in an inactive state. The rate of PARG spatiotemporal recruitment supports a second step whereby subsequent recruitment of PARG corresponds to the dissipation of PARP1 and PAR at DNA damage sites, dePARylation of Polθ, and activation of TMEJ. These studies elucidate an unprecedented mechanism of PARG activation of TMEJ and reveal the molecular basis by which PARG inhibition suppresses the DDR[41].

## Results
### PARP1 interacts with and PARylates Polθ
To determine whether PARP1 interacts with Polθ in cells, a Polθ-FLAG-HA fusion protein was expressed in 293 T cells. PARP1 was among the proteins that co-immunoprecipitated with Polθ which were identified by LC-MS/MS (Fig. 1a). Furthermore, Polθ-PARP1 interaction was predicted by STING database analysis (Fig. 1b) and validated by cross-immunoprecipitations followed by Western analysis (Fig. 1c). The Polθ-PARP1 interaction was also visualized by high resolution confocal microscopy detecting immunofluorescent co-localization in 293 T cells treated with ionizing radiation or etoposide (Fig. 1d and Supplementary Fig. 1, respectively); Polθ promotes resistance to ionizing radiation and etoposide[18,19]. As expected, Polθ-PARP1 interaction was more abundant in the irradiated BRCA1-deficient MDA-MB-436 cells (Fig. 1e). Altogether, Polθ-PARP1 interaction was detected using four experimental approaches and two cell lines.

Although multiple proteomic studies have previously identified PARP1 protein substrates, Polθ was not detected in these reports[46–49]. Thus, to determine if the Polθ-PARP1 interaction causes PARylation of Polθ, we directly investigated PARP1-mediated PARylation of recombinant human Polθ polymerase (Polθ-pol) and helicase (Polθ-hel) domains individually (Fig. 1f). Our prior studies demonstrated the respective biochemical activities of the Polθ-pol and Polθ-hel domains in vitro[1,22,24]. Here, Polθ-pol and Polθ-hel were incubated with and without recombinant PARP1 in the presence and absence of NAD+ and PARP1 activating DNA in vitro. Recombinant N-terminal His-tagged PARP1 was purified using a previously reported expression vector[50]. Next, the proteins were resolved in an SDS polyacrylamide gel in order to visualize increases in molecular weight due to PARylation. Figure 1f, lanes 3 and 6 demonstrate an increase in the molecular weights of Polθ-hel and Polθ-pol, respectively, in the presence of PARP1, NAD+ and DNA. No shift in molecular weights was observed for Polθ-hel and Polθ-pol in the absence of NAD+ and DNA (Fig. 1f, lanes 2 and 4). As a positive control for in vitro PARylation, we demonstrate a large increase in the molecular weight of PARP1 due to its auto-PARylation activity in the presence of NAD+ and activating DNA (Fig. 1f, lane 10). As negative controls, Polθ-hel and Polθ-pol proteins were resolved in the SDS gel without PARP1, NAD+ and DNA (Fig. 1f, lanes 1 and 4, respectively), and the mobility of these respective bands corresponds to their expected molecular weights without PARylation. As an additional negative control, we demonstrate that the molecular weight of BSA is not affected by pre-incubation with PARP1, NAD+ and DNA and is therefore not PARylated (Fig. 1f, lanes 7 and 8). Western blot controls further confirm that Polθ-pol and Polθ-hel domains are PARylated by PARP1 in vitro (Supplementary Fig. 2a). Although mapping PARylation sites on Polθ is beyond the scope of our study, we observed that PARP1 is unable to PARylate a mutant version of Polθ-pol lacking five disordered domains (Supplementary Fig. 2b)[51]. This suggests that PARP1 exclusively PARylates Polθ-pol within one or more disordered domains previously described[51]. We did not observe PARP1 PARylation of Polθ's central domain which is thought to be disordered due to lack of predicted secondary structures (Supplementary Fig. 2c).

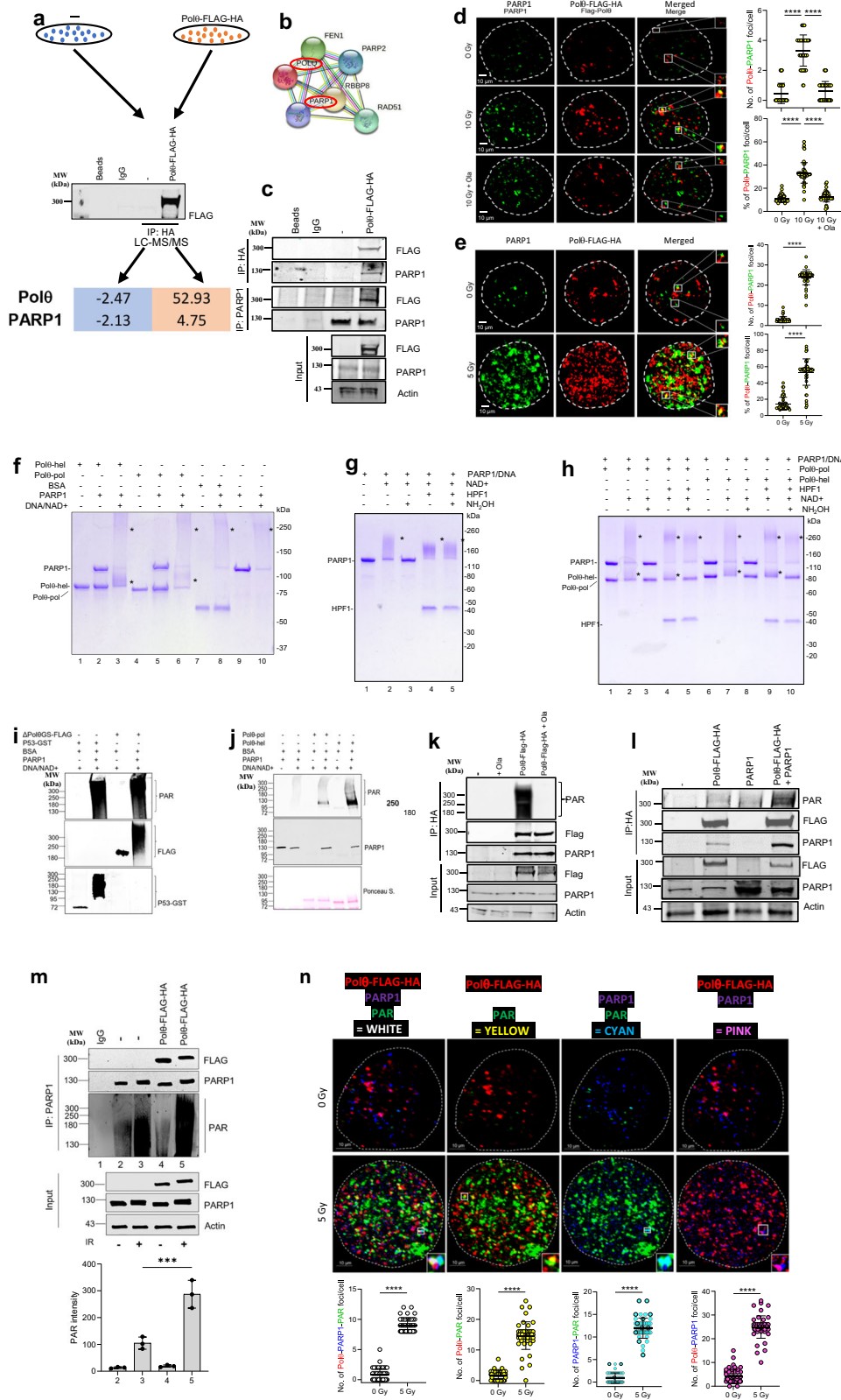

Importantly, HPF1 has been shown to regulate PARP1 PARylation specificity by shifting it from aspartate/glutamate to serine residues[36]. In concordance with a prior report[52] we showed that HPF1 regulates PARP1 by promoting auto-PARylation of serine residues which was revealed by the subsequent addition of hydroxylamine which selectively hydrolyzes PAR linked to aspartate and glutamate residues (Fig. 1g). For example, the addition of hydroxylamine following PARP1

auto-PARylation cleaved PAR linked to aspartate and glutamate residues (Fig. 1g, compare lanes 2 and 3). The addition of HPF1, however, enabled PARP1 auto-PARylation on serines, as indicated by serine-linked PAR resistance to hydrolysis by hydroxylamine (Fig. 1g, compare lanes 4 and 5). Unexpectedly, we observed that HPF1 did not regulate PARP1 PARylation of Polθ constructs (Fig. 1h). For example, although hydroxylamine hydrolyzed PARP1-mediated PAR post-

**Fig. 1 | PARP1 forms a complex with Polθ resulting in HPF1-independent Polq PARylation. a** Upper panel−293T cells expressing Polθ-FLAG-HA. Middle panel−Detection of Polθ-FLAG-HA in anti-HA Immunoprecipitates (IP) with anti-FLAG antibody. Lower panel−intensity fold change of PARP1 in anti-HA IP from 293T cells expressing Polθ-FLAG-HA assessed by LC-MS/MS analysis. **b** STING database prediction of PARP1 interactions. **c** Polθ −PARP1 interaction detected by anti-HA and anti-PARP1 IP probed with anti-FLAG (Polθ-FLAG-HA) and anti-PARP1. **d, e** Co-localization of Polθ-FLAG-HA and PARP1 in **d** 293T cells and **e** MDA-MB-436 cells overexpressing Polθ-FLAG-HA and exposed to irradiation and 1 μM olaparib using confocal microscopy. Dimentions:10 μm or 100 nm (magnified foci). Quantification of yellow foci in cells ($n = 30$) stems from three independent repeats: upper panels−mean number ± SD and lower panels -% of Polθ-FLAG-HA + PARP1 foci/cell. ****$p < 0.0001$ using one-way ANOVA (**d**) and two-tailed Student's $t$ test (**e**). **f** PARP1 PARylates Polθ-pol and Polθ-hel in vitro. SDS gel showing increase in the molecular weight (MW) of Polθ-pol and Polθ-hel in vitro by PARP1 PARylation. **g** PARP1 self-PARylation in vitro occurs in the presence of HPF1 predominantly on serine residues. SDS gel showing size increase of PARP1 due to self-PARylation. $NH_2OH$ treatment shows that serine residues of PARP1 are mainly self-PARylated in presence of HPF1. **h** HPF1 does not promote PARP1-NAD PARylation of Polθ-pol and Polθ-hel. SDS gel showing HPF1 promotes PARP1 self-PARylation on serine residues

in the presence of $NH_2OH$ which prevents PARylation on glutamate and aspartate residues **f**−**h** represent at least two experiments. * indicates shift in MW in **f**−**h**. **i, j** PARP1-mediated PARylation of recombinant purified Polθ fragments: ΔPolθGS-FLAG (polymerase-helicase domain fusion) protein (**i**), and Polθ-pol and Polθ-hel (**j**). Purified p53-GST serves as control substrate. Western blot detected PARylation of Polθ and GST-p53 proteins. **k, l** Western blot detection of PARylated Polθ in anti-HA IP from (**k**) 10 Gy irradiated (IR) 293T (−) and Polθ-FLAG-HA transfected 293T cells treated or not with 1 μM olaparib (Ola), and (**l**) 293T cells (−) and these over-expressing PARP1 and/or Polθ-FLAG-HA. The panel represents three independent experiments. **m** Western blot detection of PARylated Polθ-FLAG-HA in anti-HA IPs from IR (+) or not (−) MDA-MB-436 cells and these expressing Polθ-FLAG-HA. Bottom panel: mean ± SD of the quantitation of PARylation from three independent biological replicates. ***$p = 0.000185$ using one-way ANOVA. **n** Co-localization of Polθ-FLAG-HA, PARP1, and/or PAR in MDA-MB-436 cells expressing Polθ-FLAG-HA and exposed to IR followed by 20 min incubation. Dashed line shows nuclear border. Scale bar: 10 μm or 100 nm (magnified foci). Co-localization experiments were repeated three times with $n = 30$ cells analyzed, and representative confocal microscopy images shown. Mean number ± SD of the indicated foci formation is shown below; ****$p < 0.0001$ using two-tailed Student's $t$ test. Source data are provided as a Source Data file.

translational modifications on Polθ-pol and Polθ-hel domains (Fig. 1h, lanes 3 and 8), the addition of HPF1 during the PARP1 PARylation reaction did not generate hydroxylamine resistant serine-linked PAR chains on these Polθ constructs, which is in contrast to PARP1 (Fig. 1h, lanes 5 and 10). Hence, these data indicate that the Polθ helicase and polymerase constructs were primarily PARylated on aspartate and glutamate residues even in the presence of HPF1. We note that HPF1 is not PARylated by PARP1 compared to the Polθ constructs which serves as another selectivity control for PARylation, in addition to BSA (Fig. 1h, lanes 4 and 9). Taken together, our data reveal that HPF1 does not regulate PARP1 PARylation of Polθ enzymatic domains.

Next, PARP1-mediated PARylation of Polθ constructs was examined by Western blot analysis where recombinant human Polθ-hel, Polθ-pol, and a previously characterized 3X-FLAG-tagged Polθ polymerase-helicase fusion protein (PolθΔcen) lacking the long unstructured central domain were incubated with commercially available PARP1 in the presence of NAD+ and DNA[1]. GST-p53 protein was used as a positive control for in vitro PARylation. Western blotting with anti-PAR antibody clearly demonstrated that all of the Polθ constructs were PARylated by PARP1 (Fig. 1i, j).

To demonstrate that PARP1 PARylates Polθ in cells, 293 T cells ectopically expressing Polθ-FLAG-HA were irradiated and treated with the PARP1 inhibitor olaparib. We applied this protocol because endogenous Polθ protein is relatively not abundant in normal cells[53] which might preclude efficient detection of PARylated Polθ. Western blot analysis of anti-HA immunoprecipitates revealed robust PARylation of the immunoprecipitated Polθ-FLAG-HA, which was abrogated by olaparib (Fig. 1k). Moreover, ectopic expression of PARP1 resulted in enhanced PARylation of Polθ in 293 T cells (Fig. 1l).

To support our claim that PARP1 PARylates Polθ in vivo, 293 T cells and these ectopically expressing Polθ-FLAG-HA were irradiated or not 20 min. before protein extraction. As expected, Western blot analysis revealed that irradiation increased the detection of PARylated proteins below 250 kDa marker in anti-PARP1 immunoprecipitates (Fig. 1m, lane 3). However, abundant PARylation signal above and around 300 kDa marker was detected only in anti-PARP1 immunoprecipitates from the irradiated cells expressing Polθ-FLAG-HA, consistent with PARylation of Polθ-FLAG-HA (Fig. 1m, lane 5).

To provide additional evidence that Polθ is PARylated in cells, intracellular Polθ-FLAG-HA, PARP1, and PAR marks were immunostained in MDA-MB-436 cells before and 20 min. after 5 Gy irradiation followed by three-color confocal microscopy analysis of their co-localization. Yellow foci detected PARylated Polθ-FLAG-HA whereas white foci marked Polθ-FLAG-HA and PARylated PARP1 co-localization. Both, yellow and white foci were enhanced in the irradiated cells,

however foci representing PARylated Polθ-FLAG-HA were 1.67× more frequent than these showing co-localization of Polθ-FLAG-HA and PARylated PARP1 (Fig. 1n).

In conclusion, we present compelling evidence that PARP1 binds to and PARylates Polθ in vitro and in the cells.

## PARylation of Polθ suppresses its DNA binding and TMEJ activities

Since auto-PARylated PARP1 (PAR-PARP1) results in its dissociation from DNA[54], we envisaged that PARylation of Polθ also suppresses its DNA binding activity. For example, the addition of negatively charged PAR chains to Polθ-pol and Polθ-hel is likely to repel the negative charges along the phosphate back-bone of DNA and therefore suppress DNA binding. Indeed, using electrophoresis mobility shift assay (EMSA), we found that Polθ-hel was unable to bind ssDNA following its incubation with purified N-terminal His-tagged PARP1, NAD+ and a PARP1 activation substrate (PAS) DNA (Fig. 2a, lane 5). Omitting PARP1 or NAD+ restores Polθ-hel ssDNA binding (Fig. 2a, lanes 3 and 4). As a control, we show that PARP1 does not bind the ssDNA probe used for detecting Polθ-hel ssDNA binding (Fig. 2a, lanes 6−8). These data are consistent with prior biophysical studies showing that PAR-Polθ-hel dissociates from DNA[55]. We further found that incubation of Polθ-hel with PARP1, NAD+ and DNA suppresses its ATPase activity (Fig. 2b). Because Polθ-hel ATPase activity is strongly stimulated by ssDNA binding, these data indicate that the inability of PAR-Polθ-hel to bind ssDNA is responsible for its deficient ATPase activity. Considering that Polθ-pol is also PARylated (Fig. 1f, j), we next investigated PARP1 and NAD+ effects on ssDNA binding by the Polθ-pol:Polθ-hel fusion protein (PolθΔcen) which exhibits identical biochemical activities as full-length Polθ and was characterized in prior studies[1]. PolθΔcen was also unable to bind ssDNA in the presence of PARP and NAD+ (Fig. 2c). We further found that both Polθ-hel and Polθ-pol DNA binding activities are suppressed by the addition of recombinant PAR chains in trans (Supplementary Fig. 3), but not by the addition of ADP ribose (Supplementary Fig. 4). This indicates that PAR polyanion chains act as nucleic acid mimics that suppress DNA binding by Polθ-pol and Polθ-hel even when added in trans.

Considering that PARylated Polθ constructs were deficient in DNA binding, this indicated that PARylated Polθ constructs would not be able to perform TMEJ. To test this, we utilized an in vitro TMEJ assay that has been well documented in multiple reports (Fig. 2d)[1,22]. Here, Polθ-pol, PolθΔcen or full-length Polθ (Fl-Polθ), characterized in prior biochemical studies[1], were incubated with a radio-labelled TMEJ model substrate containing 3′ ssDNA overhangs with 6 bp microhomology (palindrome, 5′-CCCGGG-3′) along with dNTPs, standard Tris-HCl

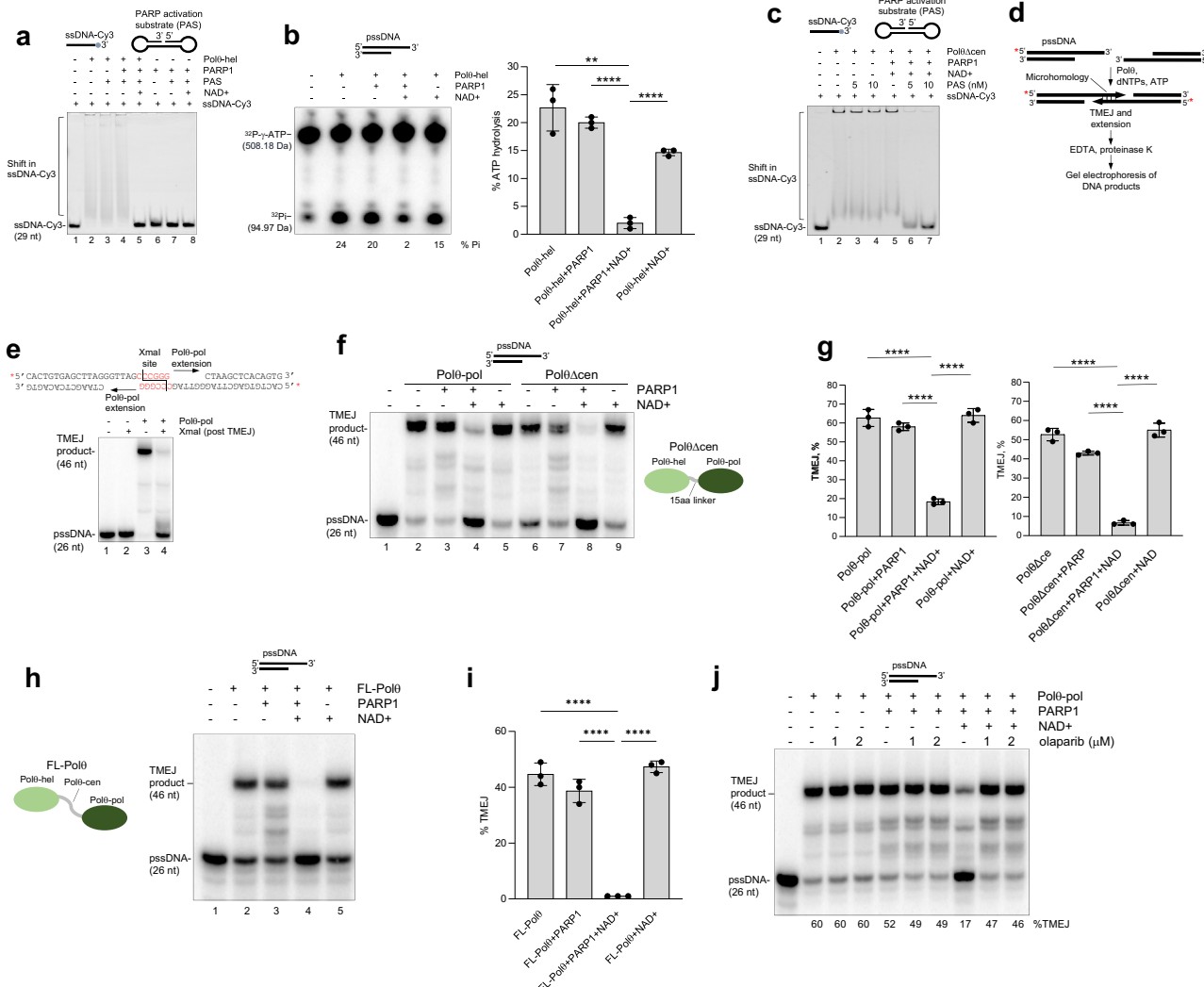

**Fig. 2 | PARylated Polθ is inhibited in DNA binding and TMEJ. a** EMSA showing that PARylated Polθ-hel is inhibited in ssDNA binding. Cy3-ssDNA probe and PARP1 activating substrate (PAS) are indicated. **b** Denaturing gel showing that the presence of PARP1 and NAD+ inhibit Polθ-hel ATPase activity (left).% inorganic phosphate (PI) formation indicated. pssDNA substrate indicated. Bar plot showing mean ± SD results from ATPase assay (right). ****$p < 0.0001$ and ***$p < 0.001$ using unpaired two-tailed $t$ test, statistics from three biological replicates (1 vs 3 $p = 0.001119$, 2 vs 3 $p = 0.000025$, 3 vs 4 $p = 0.000045$). **c** EMSA showing that the presence of PARP1 and NAD+ inhibit PolθΔcen ssDNA binding. Cy3-ssDNA probe and PARP1 activating substrate (PAS) are indicated. **d** Schematic of in vitro TMEJ assay using pssDNA substrates containing 3′ terminal microhomology. **e** Denaturing gel showing XmaI cleavage of expected TMEJ product. Schematic of pssDNA substrate and TMEJ synapse with XmaI recognition site indicated in red

(top). **f** Denaturing gel showing that the presence of PARP1 and NAD+ suppress Polθ-pol and PolθΔcen TMEJ activities. **g** Bar plots showing mean% TMEJ activity ± SD by Polθ-pol and PolθΔcen in the presence of the indicated factors. ****$p < 0.0001$ using unpaired two-tailed $t$-test, statistics from three biological replicates (1 vs 3 $p = 0.00002$, 2 vs 3 $p = 0.00001$, 3 vs 4 $p = 0.000025$, 5 vs 7 $p = 0.000086$, 6 vs 7 $p = 0.000011$, 7 vs 8 $p = 0.000035$). **h** Denaturing gel showing that the presence of PARP1 and NAD+ suppress FL-Polθ TMEJ activity. **i** Bar plots showing mean ± SD results from%TMEJ activity by FL-Polθ and in the presence of the indicated factors. ****$p < 0.0001$ using unpaired two-tailed $t$-test, statistics from three biological replicates (1 vs 3 $p = 0.000048$, 2 vs 3 $p < 0.00001$, 3 vs 4 $p < 0.00001$). **j** Denaturing gel showing that olaparib prevents suppression of Polθ-pol TMEJ in the presence of PARP1 and NAD+. Experiments **a**, **c**, **e**, **j** were repeated at least two times with similar results. Source data are provided as a Source Data file.

buffer along with MgCl₂ in the presence or absence of PARP1 and NAD +. A previously developed positive control for in vitro TMEJ using XmaI endonuclease to digest the expected TMEJ product is shown in Fig. 2e[1]. As predicted, all three Polθ constructs were inhibited in TMEJ in the presence of PARP1 and NAD+, and the omission of PARP1 or NAD+ restored TMEJ activity (Fig. 2f–i). The presence of PARP1 alone showed minor but insignificant suppression of TMEJ activity likely due to its competitive binding to the double-strand/ssDNA junction (Fig. 2g, i). The addition of olaparib also restored Polθ-pol TMEJ activity as expected (Fig. 2j).

Additional Western blots show that the previously characterized PARylation-deficient/MARylation-proficient PARP1(E988Q) mutant MARylates Polθ-pol and Polθ-hel domains as expected (Supplementary

Fig. 5a)[56]. We show that MARylation of Polθ-pol by PARP1(E988Q) only partially inhibits its TMEJ activity (Supplementary Fig. 5b). In addition, only relatively low concentrations of NAD⁺ (<20 μM) are needed for wild-type PARP1 + NAD⁺ to suppress Polθ-pol TMEJ (Supplementary Fig. 5c). Hence, PARP1-mediated PARylation, but not MARylation, of Polθ-pol plays a major role in suppressing its TMEJ activity.

## PARP1 and PARG are essential for TMEJ in vitro and in cells
The data in Fig. 2 presented a conundrum in Polθ-PARP1 biology. For example, multiple studies found that suppression or inhibition of PARP1 reduces the recruitment of Polθ to DNA damage foci[2,21]. Furthermore, additional studies have shown that PARP1 promotes Alt-EJ[57,58], and we confirm that PARP1 promotes TMEJ activity in cells

(Supplementary Fig. 6). Thus, current models propose that PARP1 promotes TMEJ by facilitating Polθ recruitment to DNA damage. Yet our in vitro findings directly show that PAR-Polθ is defective in DNA binding and end-joining. Considering that prior cellular reports did not evaluate the effects of PARG in their Polθ and TMEJ studies, the current models of how PARP1 influences Polθ and TMEJ are likely to be incomplete. Because PARG functions to dePARylate proteins, we envisaged that PARG would restore Polθ-mediated end-joining activity by removing the repressive PAR marks.

Indeed, a time course of Polθ-pol TMEJ in vitro demonstrates that the addition of PARG restores its TMEJ activity in the presence of PARP1 and NAD+ (Fig. 3a). In contrast, the addition of PARP1 and NAD+ without PARG shows a significant reduction in the relative rate of TMEJ compared to Polθ-pol alone or Polθ-pol with PARP1 but without NAD+ as expected. Here again, we observe a minor but insignificant reduction in Polθ-pol TMEJ activity when PARP1 is added without NAD+, likely due to competitive PARP1 binding to the ssDNA/dsDNA junction. Consistent with these results, we observe that PARG also restores the TMEJ activity of FL-Polθ and PolΔcen in the presence of PARP1 and NAD+ in vitro (Fig. 3b). As a control, we show that PARG does not affect the respective TMEJ activity of the three constructs of Polθ in the absence of PARP1 and NAD+ (Fig. 3b, c). Finally, we found that HPF1 had no effect on PARP1-NAD⁺ PARylation-dependent suppression of Polθ TMEJ activity in vitro (Fig. 3d), which is consistent with our observation that HPF1 does not regulate PARP1 PARylation of Polθ constructs (Fig. 1h). Taken together, our results indicate that PARG-mediated dePARylation of PAR-Polθ reactivates its TMEJ activity likely by restoring its capacity to bind DNA.

Since PARG counteracts PARP1-mediated PARylation and the inhibitory effects of PARylation of Polθ on TMEJ in vitro, we probed whether PARG is necessary for TMEJ in cells. Here, we utilized a previously characterized small-molecule inhibitor of PARG[59], and a U2OS cellular model of TMEJ where an ectopically expressed I-SceI endonuclease induces a DSB in a single copy of genome-integrated EJ2-GFP reporter cassette which requires Polθ to repair the DSB using 8-bp microhomology (Supplementary Fig. 6d)[60]. As expected, downregulation of PARP1 by shRNA (Fig. 3e) reduced TMEJ activity by approximately 3-fold (Fig. 3f). Remarkably, downregulation of PARG (Fig. 3e) exerted a similar inhibitory effect (Fig. 3f) demonstrating that PARG is also essential for TMEJ in cells. To confirm these findings, the cells were treated with PARG inhibitor PDD00017273 or PARP1 inhibitor AZD5305 to enhance or inhibit PARylation, respectively (Fig. 3g). Consistent with the results obtained with the use of shRNAs, olaparib abrogated TMEJ activity, and the PARG inhibitor exerted a similar inhibitory effect again demonstrating that PARG is also essential for TMEJ (Fig. 3h).

HPF1, a PARP1 co-factor plays a fundamental regulatory role in PARP1-mediated PARylation by changing the PARylation specificity of from aspartate/glutamate to serine[36,61]. Therefore, we compared TMEJ activity in U2OS cells and in cells where HPF1 was downregulated by siRNA (Fig. 3i). Results presented in Fig. 3j clearly show that HPF1 does not affect TMEJ.

We investigated the impact of PARP1 and PARG inhibitors on Polθ recruitment to irradiation-induced DNA damage foci marked by γH2AX. U2OS cells expressing Polθ-FLAG-HA were irradiated or not followed by detection of nuclear foci of Polθ-FLAG-HA and γH2AX. The co-localization of Polθ-FLAG-HA foci and γH2AX increased about 5-fold at 120 min. post-irradiation, and both PARP1i and PARGi abrogated the co-localization (Fig. 3k). We next examined the interaction of Polθ and γH2AX in chromatin extracts following inhibition of PARP1 and PARG. Anti-γH2AX chromatin immunoprecipitations revealed abundant increase of Polθ-FLAG-HA 120 min. after irradiation whereas PARP1i and PARGi abrogated this effect (Fig. 3l).

Taken together, the data presented in Fig. 3 suggested an essential temporal roles for both PARP1 and PARG in TMEJ.

## PARP1 and PARG regulate spatiotemporal recruitment of Polθ to intracellular DNA damage

Considering that PAR-Polθ is deficient in TMEJ, and that PARG reactivates Polθ end-joining by removing PARylation marks, it would be expected that PARG activates TMEJ subsequent to PARP1-mediated PARylation of Polθ in cells. Thus, we next investigated the spatial-temporal recruitment of PARP1, PARG, and Polθ at DNA damage foci. Here, U2OS cells expressing Polθ-FLAG-HA were irradiated followed by time-dependent detection of nuclear foci of Polθ-FLAG-HA, PARP1, PARG and PAR. The intensity of Polθ-FLAG-HA foci increased about threefold at 20 min post-irradiation and decreased only modestly at 60 and 120 min. (Fig. 4a). Conversely, PARP1, PARG and PAR foci displayed dramatic kinetic changes. PARP1 and PAR foci rose by four- to sixfold at 20 min and then continuously dissipated at 60 and 120 min after irradiation; at the same times PARG foci demonstrated a constant increase (Fig. 4a). These data suggest rapid PARP1 recruitment to DNA damage along with Polθ followed by subsequent PARG recruitment which corresponds to PARP1 dissipation while Polθ remains at DNA damage foci.

Next, we examined co-localization of Polθ, PARP1 and PARG, as well as PARylation of Polθ after irradiation. Immunofluorescence studies using high-resolution confocal microscopy detected a cohort of Polθ-FLAG-HA foci co-localized with PARP1 and PAR foci at 20 min after irradiation, and the co-localizations sharply decreased at 60 and 120 min (Fig. 4b). According to the results in Fig. 1n ~60% of Polθ-FLAG-HA + PAR foci detected by dual-color immunofluorescence represented PARylated Polθ-FLAG-HA and about 40% resulted from the interaction of PARylated PARP1 with Polθ-FLAG-HA. We also observed a steady increase of Polθ-FLAG-HA foci co-localized with PARG foci at 20, 60 and 120 min. post-irradiation. To further investigate these potential cellular interactions, we examined the immunoprecipitates of Polθ-FLAG-HA for the presence of PARP1 and PARG, and PARylation of Polθ. Consistent with the immunofluorescence results, detection of PARylated Polθ (PARylation detected above and ~300 kDa marker, Fig. 1m) and PARP1 interaction with Polθ was significantly higher in anti-HA immunoprecipitates obtained after 20 and 60 min post-irradiation, however, this interaction was reduced at 120 min (Fig. 4c). Conversely, detection of PARG interaction with Polθ steadily increased during these time points. In conclusion, the magnitude of co-immunoprecipitation of Polθ with PARP1 and PARG affected the PARylation status of Polθ.

Thus, we next investigated post-irradiation temporal recruitment of Polθ to DNA damage marked by γH2AX, and how this corresponded to PARP1 and PARG recruitment in wild-type and *HPF1* KO U2OS cells (Fig. 5a). Anti-HA immunoprecipitations from chromatin fraction showed time-dependent increasing localization of Polθ-FLAG-HA and γH2AX on chromatin in irradiated U2OS wild-type and HPF1 KO cells (Fig. 5b). This effect was associated with dePARylation of Polθ-FLAG-HA which was accompanied by growing detection of PARG and diminishing presence of PARP1. Temporal accumulation of dePARylated Polθ-FLAG-HA and PARG accompanied by dissipation of PARP1 was also detected in anti-γH2AX immunoprecipitates from the chromatin extracts of U2OS cells (Fig. 5c).

Immunofluorescent studies corroborated the results from chromatin extracts and revealed time-dependent increased co-localization of Polθ-FLAG-HA and γH2AX in irradiated U2OS wild-type and HPF1 KO cells (Fig. 5d). This effect correlated with enhanced co-localization of γH2AX and PARG but decreased co-localization of γH2AX and PARP1 (Fig. 5e). As a control to confirm that endogenous Polθ behaves similarly to ectopically expressed Polθ-FLAG-HA, we employed murine 32Dcl3 cells expressing high levels of Polθ due to malignant transformation by oncogenic tyrosine kinases such as FLT3(ITD), JAK2(V617F) and BCR-ABL1[20]. In concordance with the studies of Polθ-FLAG-HA, we found that accumulation of chromatin-bound endogenous Polθ was preceded

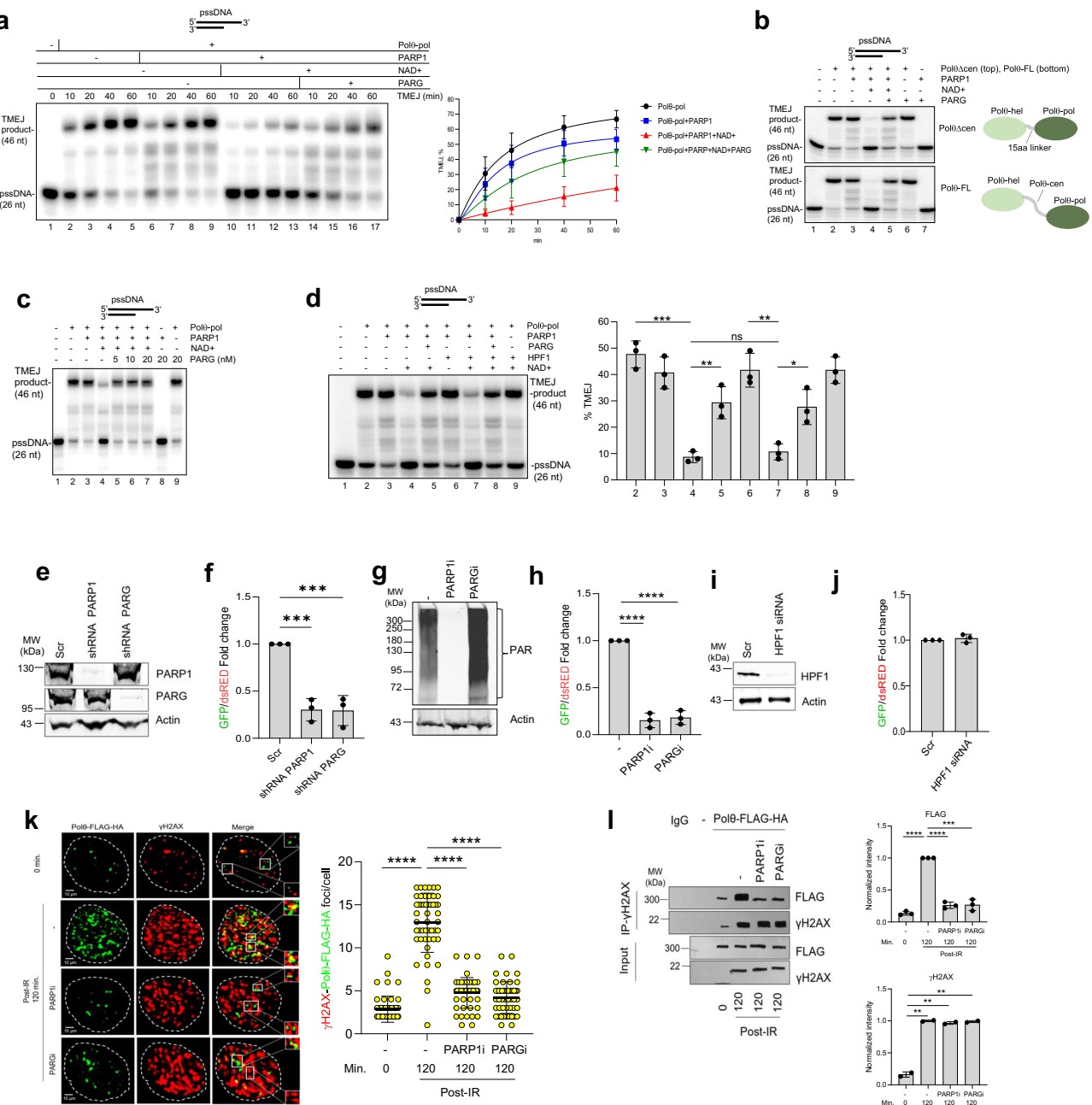

**Fig. 3 | PARP1 and PARG are essential for TMEJ. a** Denaturing gel showing a time course of Polθ-pol TMEJ in the presence and absence of PARP1, NAD+, and PARG (left). Scatter plot showing a time course of Polθ-pol TMEJ in the presence and absence of PARP1, NAD+, and PARG (right); data represent mean ± SD results from three biological replicates. **b** Denaturing gels showing that PARG prevents inhibition of PolθΔcen and FL-Polθ TMEJ by PARP1 + NAD+. **c** Denaturing gel showing that PARG prevents inhibition of Polθ-pol TMEJ by PARP1 + NAD+. **d** Left: denaturing gel showing that the presence of HPF1 does not affect PARP1-NAD suppression or PARG rescue of Polθ-pol TMEJ activity in vitro. Right: Bar plot showing% MMEJ activity by Polθ-pol (pol) and in the presence of the indicated factors. Data represent mean ± SD from three biological replicates; ***$p < 0.001$, **$p < 0.01$, and *$p < 0.05$ using unpaired two-tailed *t*-test (2 vs 4 $p = 0.0003$, 4 vs 5 $p = 0.0052$, 4 vs 7 $p = 0.402$, 6 vs 7 $p = 0.0017$, 7 vs 8 $p = 0.0159$). Experiments 3b–c were repeated two or three times with similar results. **e, f** Western analysis of the expression of PARP1 and PARG (**e**) and TMEJ activity: mean ± SD ratio of GFP + /dsRED+ cells from three biological replicates (**f**) in U2OS-EJ2-GFP cells transduced with the indicated shRNAs; ***$p < 0.001$ ($p$ value 1 vs 2 $p = 0.0006$, 1 vs 3 $p = 0.0005$) using one-way

ANOVA. **g, h** Western analysis of PARylation (**g**) and TMEJ activity assay: mean ± SD ratio of GFP + /dsRED+ cells from three biological replicates (**h**) in U2OS-EJ2-GFP cells treated with the indicated inhibitors; ****$p < 0.0001$ using one-way ANOVA ($p$ value 1 vs 2 $p < 0.00001$, 1 vs 3 $p < 0.00001$). **i, j** Western analysis of HPF1 and (**i**) and TMEJ activity: mean ± SD ratio of GFP + /dsRED+ cells from three biological replicates (**j**) in U2OS-EJ2-GFP cells transduced with the indicated siRNAs. **k, l** U2OS cells (C) and these expressing Polθ-FLAG-HA were analyzed 120 min post-10 Gy irradiation. Irradiated cells were treated with PARP1i or PARGi. **k** Left: Polθ-FLAG-HA and γH2AX foci formation and co-localization. Dimentions:10 μm or 100 nm (magnified foci). *Right*: mean ± SD of Polθ-FLAG-HA - γH2AX yellow foci co-localization in individual nuclei ($n = 50$) from two independent repeats. ****$p < 0.0001$ using one-way ANOVA. ($p$ value 1 vs 2 $p < 0.00001$, 2 vs 3 $p < 0.00001$, 2 vs 4 $p < 0.000032$). **l** Left: western blot analysis of anti-γH2AX immunoprecipitates from chromatin extracts from three independent biological replicates. Right: mean ± SD of protein band intensity is shown. ****$p < 0.0001$, ***$p < 0.001$, and **$p < 0.01$ using one-way ANOVA. Source data are provided as a Source Data file.

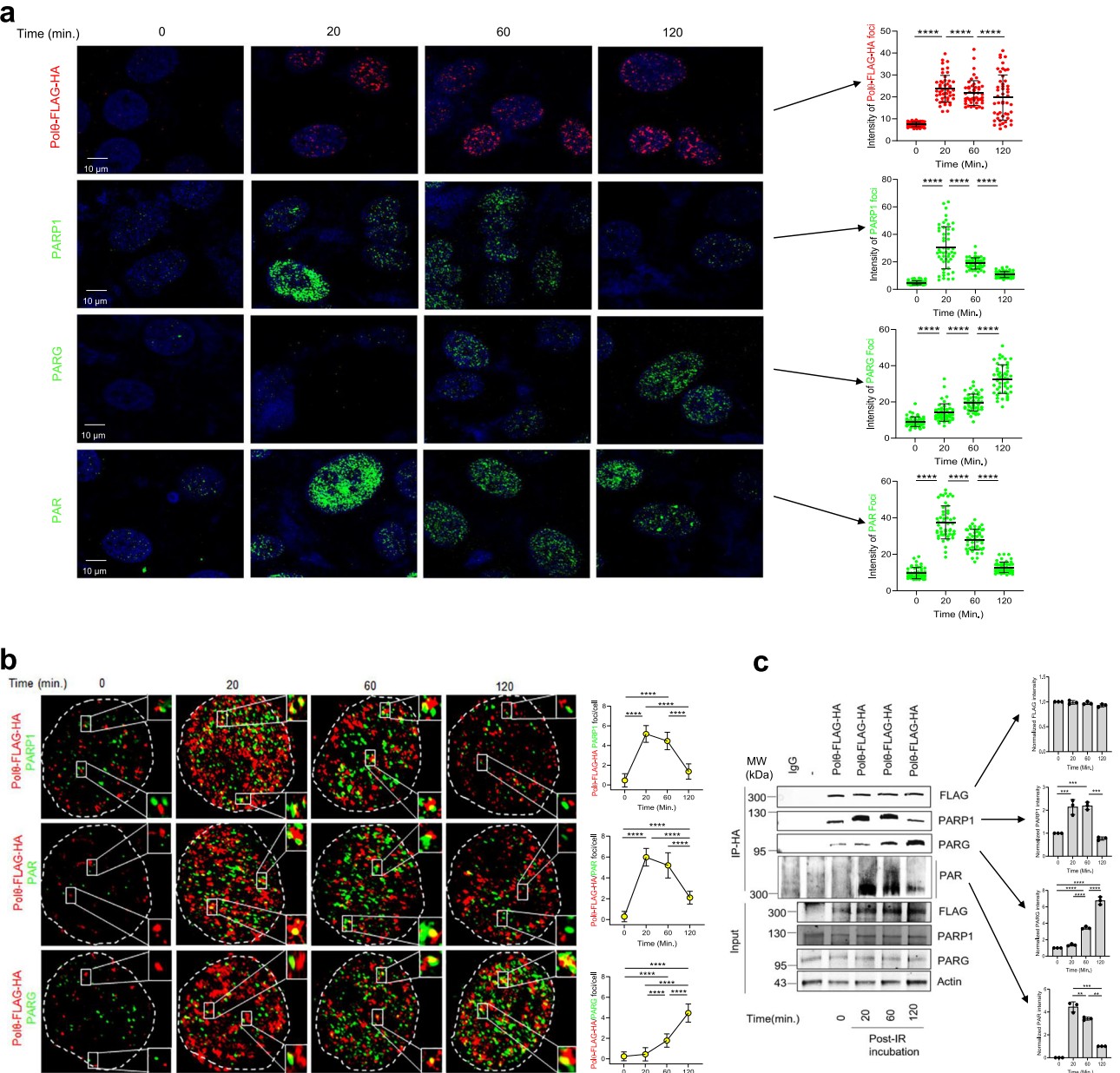

**Fig. 4 | Spatial and temporal interaction of Polθ, PARP1, and PARG after DNA damage.** U2OS cells were irradiated with 10 Gy at time 0 followed by incubation for 20, 60, and 120 min. **a** Left: panels show representative Polθ-FLAG-HA, PARP1, PARG, and PAR nuclei (*n* = 45) foci formation after irradiation from two independent biological replicates. Dimentions:10 μm or 100 nm (for magnified foci). Right - Dot-plots illustrating the mean arbitrary unit intensity ± SD of the indicated foci in individual nuclei; ****p < 0.0001 using one-way ANOVA. **b** Left: Polθ-FLAG-HA, PARP1, PARG and PAR foci formation and co-localization in U2OS cells after 10 Gy irradiation. Dimentions:10 μm or 100 nm (for magnified foci). Right: mean ± SD of Polθ-FLAG-HA–PARP1, Polθ-FLAG-HA–PARG, and Polθ-FLAG-HA–PAR yellow foci co-localization in individual nuclei (*n* = 50) from two independent biological replicates. ****p < 0.0001 using one-way ANOVA **c** Left: Western blot analysis of anti-HA immunoprecipitates obtained post-10 Gy irradiation from total cell extracts of U2OS-EJ2-GFP cells (−) and these expressing Polθ-FLAG-HA from three independent biological replicates. Right: mean ± SD of protein band intensity is shown; **p < 0.01 and ***p < 0.001 using one-way ANOVA. Source data are provided as a Source Data file.

by a sharp increase of PARP1 at 20 min post-irradiation followed by a significant reduction at 120 min, and this late time point corresponded to retention of Polθ and the highest levels of PARG detection (Supplementary Fig. 7).

Taken together, the data presented in Figs. 4 and 5 demonstrate that Polθ recruitment to DNA damage follows PARP1 and PAR, which supports a model in which PARP1 rapidly promotes the recruitment of Polθ to the vicinity of DNA damage via a PARylation-dependent mechanism. This effect does not depend on PARP1 co-factor, HPF1. Our findings also reveal that subsequent PARG recruitment to DNA damage corresponds with PARP1 and PAR dissipation, while

dePARylated Polθ remains in the vicinity of DNA damage where it is expected to promote TMEJ.

## PARP1 recruits Polθ to DNA via liquid demixing

Importantly, PARP1-dependent PARylation induces liquid demixing (LLPS), resulting in sequestration of PARylated and PAR binding proteins into transient and fully reversible spatially confined membrane-less compartments, including these at/near DNA damage sites[42,62,63]. Since PARP1 facilitates Polθ PARylation and its recruitment to DNA damage in cells, we hypothesized that PARP1 promotes Polθ recruitment to the vicinity of DNA via PARylation-dependent formation of

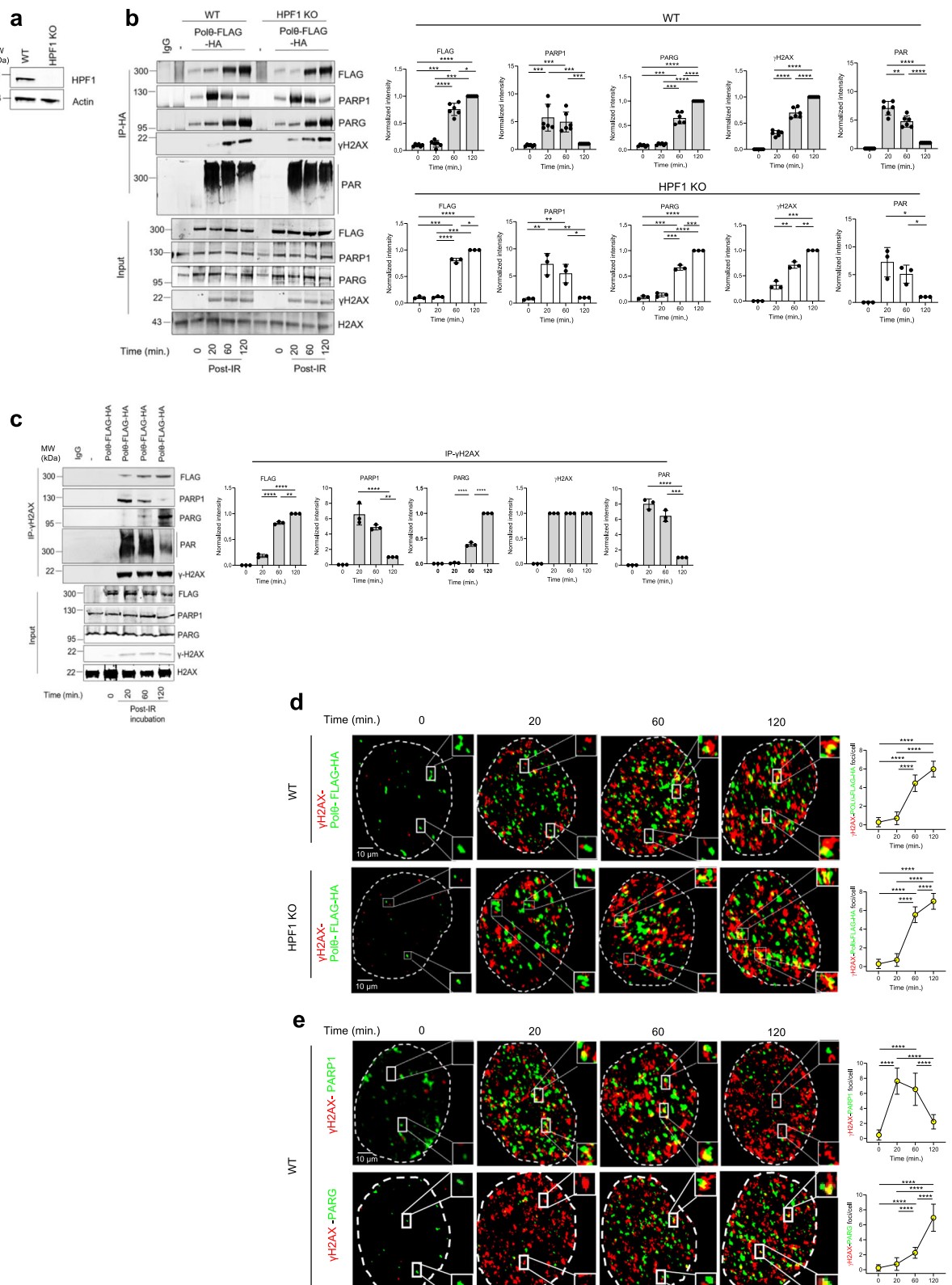

biomolecular liquid condensates. To directly test this mechanism, we examined in vitro formation of liquid condensates by PARP1 and NAD$^+$ in the presence of DNA and Polθ (Fig. 6).

Here, a Cy3-conjugated MMEJ DNA model substrate (Cy3-DNA) was used to activate PARP1 and serve as a fluorescent marker to visualize the formation of liquid droplets containing the DNA via confocal microscopy. Additionally, a red-fluorescent conjugated

Polθ-pol (Red-Polθ-pol) was visualized using the same emission and absorption spectra for Cy5. The circularity or spherical nature of fluorescent particles is widely used as an indicator for the formation of liquid droplets via LLPS[64,65]. Thus, we used particle circularity measurement as an indicator of liquid droplet formation and scored particles >1.5 micron$^2$ in size. As a positive control for liquid droplet formation via LLPS, we first examined whether mixing PARP1 and

**Fig. 5 | Spatial and temporal recruitment of Polθ, PARP1, and PARG to DNA damage sites. a** Western blot validates HPF1 knockout in U2OS *HPF1* KO cells. **b** Left: western blot analysis of anti-HA and anti-γH2AX immunoprecipitates obtained post-10 Gy irradiation from chromatin extracts of U2OS WT (−) and *HPF1*KO cells (−) and these expressing Polθ-FLAG-HA. Right: mean ± SD of protein bands intensity quantification from three independent biological replicates is shown; *$p < 0.05$, **$p < 0.01$, ***$p < 0.001$, and ****$p < 0.0001$ using one-way ANOVA. **c** Left: western blot analysis of anti-γH2AX immunoprecipitates obtained post-10 Gy irradiation from chromatin extracts of U2OS WT cells (−) and these expressing Polθ-FLAG-HA. Right: mean ± SD of protein bands intensity quantification from three independent biological replicates is shown; *$p < 0.05$, **$p < 0.01$, and ***$p < 0.001$

using one-way ANOVA. **d** Left: Post-10 Gy irradiation co-localization of γH2AX with Polθ-FLAG-HA in U2OS WT and *HPF1*KO cells expressing Polθ-FLAG-HA. Dimentions:10 μm or 100 nm (for magnified foci). Right: quantification of mean number ± SD of γH2AX-Flag - Polθ yellow foci formation ($n = 50$ cells) from two independent biological replicates is shown; ****$p < 0.0001$ using one-way ANOVA. **e** Left: Post-10 Gy irradiation co-localization of γH2AX with PARP1 and PARG in U2OS cells expressing Polθ-FLAG-HA. Dimentions:10 μm or 100 nm (for magnified foci). Right: Quantification of mean number ± SD of γH2AX−PARP1 and γH2AX−PARG yellow foci formation ($n = 50$) from two independent biological replicates is shown; ****$p < 0.0001$ using one-way ANOVA. Source data are provided as a Source Data file.

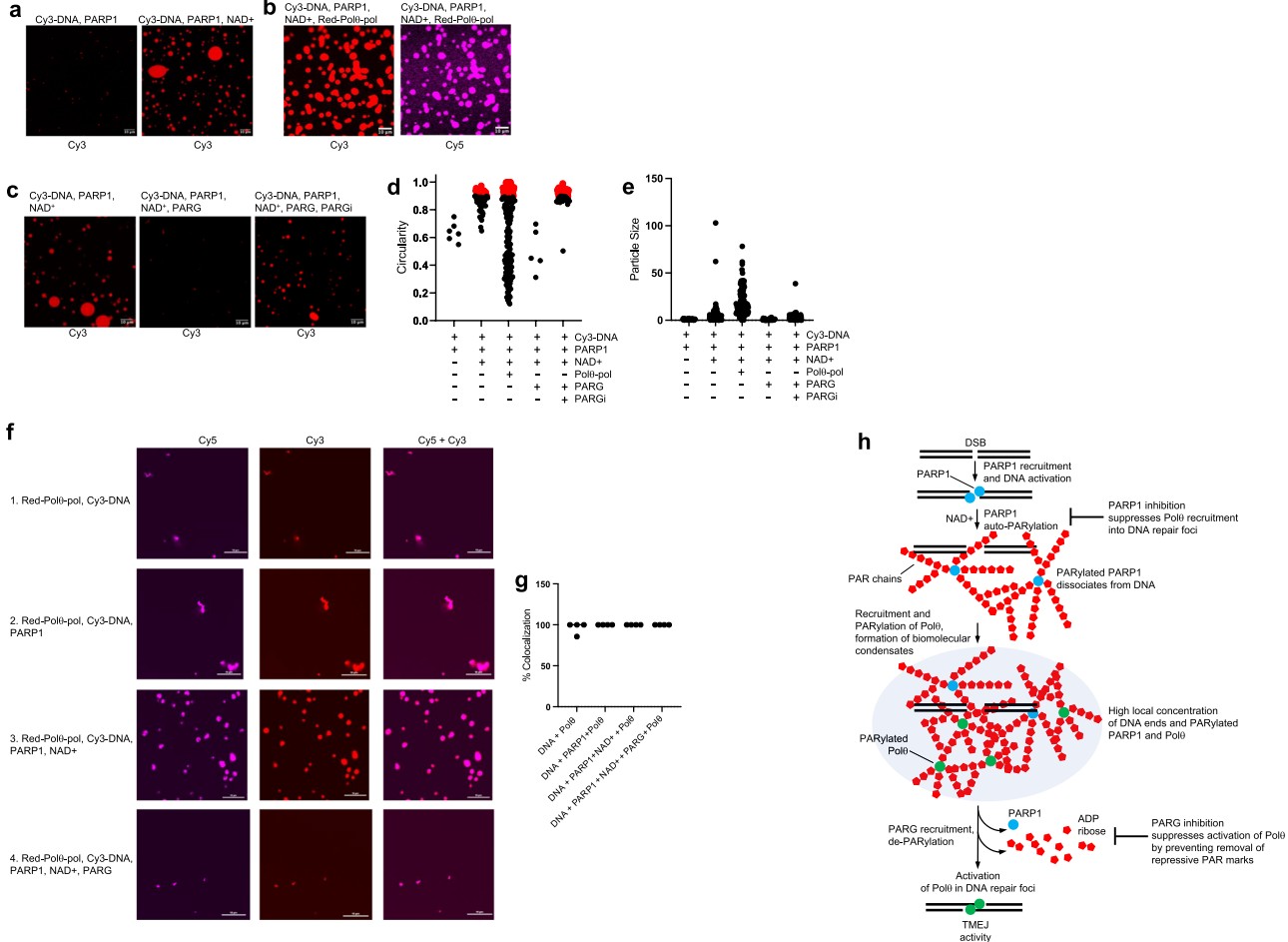

**Fig. 6 | PARP1 and PARG regulate biomolecular condensates containing Polθ and DNA. a−c** Confocal images showing that PARP1 + NAD+ promote biomolecular condensates containing Cy3-DNA and Red-Polθ-pol. Confocal microscopy images are shown of Red-Polθ-pol and Cy3-DNA under the indicated conditions following Cy3 and Cy5 fluorescence as indicated. 10 μm bars are indicated. **d, e** Plots showing circularity (**d**) and size (**e**) of particles formed under the indicated conditions. Red circles represent particles with circularity scores >0.9 (**d**). **f** Confocal images showing that Red-Polθ-pol colocalizes with Cy3-DNA in droplets formed by

PARP1 + NAD+. Confocal microscopy images are shown of Red-Polθ-pol and Cy3-DNA under the indicated conditions[1–4] following Cy3 and Cy5 fluorescence as indicated. 10 μm bars are indicated. **g** Quantitation of the number of co-localization events between Red-Polθ-pol and Cy3-DNA under the indicated conditions. Co-localization events from 4 separate confocal microscopy images are quantitated. **h** Model of Polθ TMEJ regulation by PARP1 and PARG. Experiments in 6**a−c**, **f** were performed at least twice. Source data are provided as a Source Data file.

NAD+ with the Cy3-DNA substrate promoted liquid droplets in vitro via confocal microscopy. Incubating PARP1 with Cy3-DNA in the absence of NAD+ did not result in droplets as expected (Fig. 6a, left). For example, only particles with a circularity score <0.9 were observed (Fig. 6d). The average size of the irregularly shaped particles observed under these conditions without NAD+ was relatively small (1.78 micron²), suggesting they represent PARP1 protein aggregates bound to Cy3-DNA (Fig. 6e). We then repeated the experiment but added NAD+ which resulted in large droplets and

78.4% of particles exhibited 0.9-1.0 circularity score (Fig. 6a, right; Fig. 6d). Particle sizes observed under this condition with NAD+ was also substantially larger which is consistent with liquid droplet formation (Fig. 6e). Next, PARP1 was mixed with Cy3-DNA, NAD+ and Red-Polθ-pol. Again, droplets were clearly observed with 37% of particles exhibiting a circularity score ≥0.9, and the observation of large particle sizes (Fig. 6b, left; Fig. 6d, e). We also observed a high degree of co-localization of Red-Polθ-pol with Cy3-DNA under these conditions which formed droplets (Fig. 6b, right; 6 f,g). Since

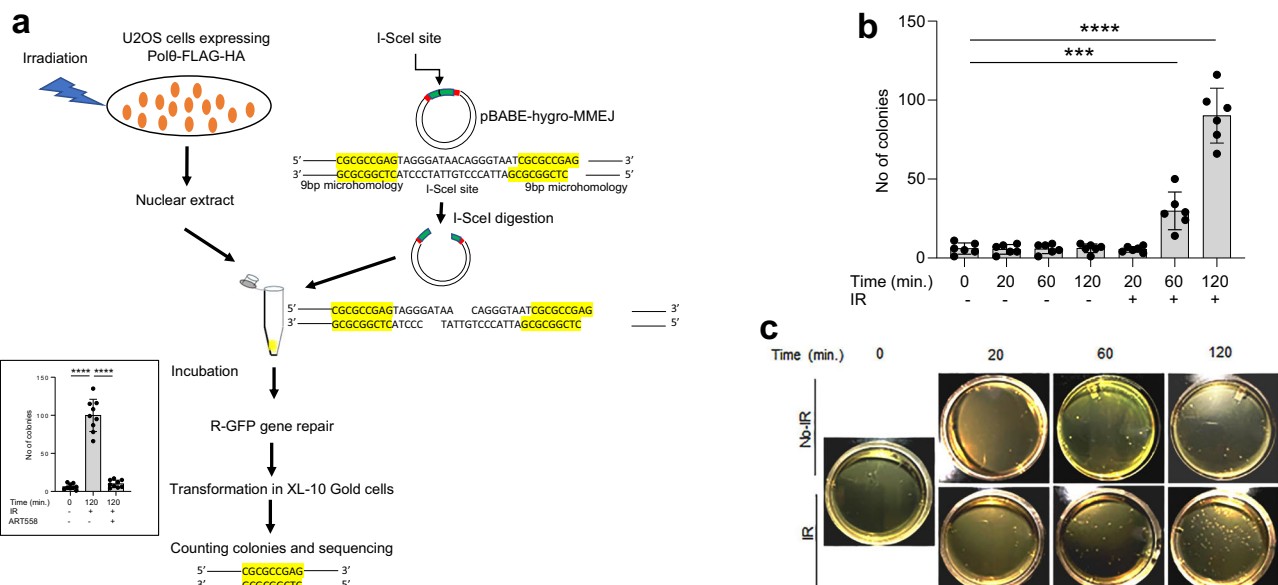

**Fig. 7 | Temporal TMEJ activity in irradiated cells. a** Schematic outline of TMEJ in vitro assay. Insert: Polθ inhibitor ART558 abrogated TMEJ activity (cells were treated with 12.5 µM ART558 for 16 h before IR and during incubation procedure) in duplicates from three independent biological replicates. ****$p < 0.0001$ using one-way ANOVA ($p$ value 1 vs 2 $p < 0.00001$, 2 vs 3 $p < 0.00001$). **b** Mean number ± SD of XL-10 Gold colonies obtained from the in vitro TMEJ assay using nuclear lysates

from U2OS cells obtained at 0, 20, 60 and 120 min. after 10 Gy irradiation (IR) or without IR from two independent experiments; ***$p < 0.001$ and ****$p < 0.0001$ using one-way ANOVA ($p$ value 1 vs 6 $p = 0.00012$, 1 vs 7 $p < 0.00001$). **c** Representative images of XL-10 Gold colonies harboring the repaired pBABE-hygro-MMEJ plasmid. Source data are provided as a Source Data file.

PARylated proteins are known to be recruited to liquid droplets via LLPS, our data indicate that Polθ-pol is recruited to liquid droplets via PARP1-NAD⁺ PARylation. As a control, we demonstrate that incubation of PARG with PARP1, Cy3-DNA and NAD⁺ prevents the formation of liquid droplets as indicated by no observable particles with a ≥ 0.9 circularity score (Fig. 6c, compare left and center panels; Fig. 6d). The addition of PARGi, however, reverses this effect, and 96% particles with ≥ 0.9 circularity score were observed (Fig. 6c, right; Fig. 6d). Hence, these data demonstrate that PARG hydrolysis of PAR chains prevents droplet formation via LLPS. Figure 6f, g quantitates the co-localization of Red-Polθ-pol and Cy3-DNA under conditions with and without PARP, NAD⁺ and PARG and shows a high degree of co-localization under all conditions. Red-Polq-pol shows a high degree of co-localization with Cy3-DNA even under conditions in which LLPS is induced by PARP1-NAD⁺. For example, the addition of NAD⁺ to Red-Polθ-pol, PARP1 and Cy3-DNA resulted in a large number of droplets in which a high degree of Red-Polθ-pol:Cy3-DNA co-localization was observed (Fig. 6f, g). Consistent with the data above, the addition of PARG significantly reduced the number of droplets which is consistent with the requirement for PAR chains to induce droplet formation via LLPS (Fig. 6f). The small number of irregular shaped elongated particles observed in the absence of NAD⁺ likely represent Red-Polθ-pol aggregates bound to Cy3-DNA, similar to above (Fig. 6f).

These results in conjunction with our findings above support a model whereby PARP1-mediated PARylation of Polθ enables its recruitment to the vicinity of DNA via LLPS (Fig. 6h). However, PAR-Polθ is unable to bind DNA and perform TMEJ. Therefore, PARG not only activates Polθ end-joining activity by removing repressive PAR marks, but also enables dissolution of PAR-dependent biomolecular condensates at DNA breaks.

### Temporal recruitment of PARG to DNA damage corresponds with cellular TMEJ activity

Finally, considering that PARG is recruited to DNA damage relatively late after PARP1 and is essential for TMEJ activity in cells, we

investigated whether the time-dependent recruitment of PARG to DNA damage corresponded with TMEJ activity in cells. We used a linearized plasmid as a DNA damage substrate and nuclear extracts as a source of DNA repair proteins similar to prior studies[66]. More specifically, a TMEJ assay was performed using the I-SceI -linearized pBABE-hygo-MMEJ reporter plasmid containing 9-bp micro-homology near the DSB and protein nuclear lysates obtained from U2OS cells prior to or following 20, 60 and 120 min after irradiation to probe the temporal effect on TMEJ (Fig. 7a). The linearized plas-mid was incubated with nuclear lysates, and the reaction mixes were used to transform bacteria; bacterial colonies containing circular-ized plasmid were counted, and the repaired plasmid was sequenced. The TMEJ specificity of the assay was confirmed by abrogation of the repair activity in nuclear lysates obtained from cells 120 min post-irradiation and treated with a previously char-acterized Polθ-pol inhibitor ART558 (Fig. 7a, insert), and by sequencing of the circularized repaired plasmids to detect the 9-bp microhomology (all sequenced colonies contained plasmids with 9-bp microhomology). TMEJ was not activated in the lysates obtained 20 min after irradiation, but it was enhanced by approxi-mately 6× and 15× in the lysates obtained after 60- and 120-min post-irradiation, respectively (Fig. 7b, c), which correlated with enhanced association of dePARylated Polθ with chromatin and γH2AX (Fig. 5b–d). Altogether, interaction of Polθ-FLAG-HA with PARP1 and its PARylation displayed opposite recruitment kinetics to that of Polθ-FLAG-HA and PARG. For example, at 20 min post-irradiation we observed an increase in Polθ-PARP1 interaction accompanied by abundant PARylation of Polθ, and this was not associated with increased detection of Polθ on chromatin and γH2AX (Fig. 5b–d) and activation of TMEJ (Fig. 7b). Subsequent accumulation of Polθ-PARG complexes at 60- and 120-min, however, was accompanied by gra-dual dePARylation of Polθ, and an increase in its detection on chromatin/γH2AX as well as elevated TMEJ activity. Taken together, these spatial-temporal studies of Polθ, PARP1, PARG, PAR and TMEJ activity in cells support our biochemical findings that PARG is essential for TMEJ via dePARylation of Polθ.

## Discussion

Using intracellular and biochemical approaches we show here that both PARP1 and PARG are required for TMEJ despite their counteracting enzymatic activities, which supports the notion that PARP1 and PARG act in the same linear pathway to promote TMEJ. We also pinpointed a unique dynamic spatiotemporal interplay between Polθ, PARP1 and PARG to regulate TMEJ activity. During the initial stage, PARP1 activated by DNA damage PARylates Polθ and facilitates its recruitment to the vicinity of DNA damage in concordance with other reports[2,21]. The mechanism by which PARP1 facilitates recruitment of Polθ to DNA breaks, however, has remained unclear. Our in vitro data indicate that PARP1-dependent PARylation facilitates recruitment of PAR-Polθ to the vicinity of DNA ends via the formation of liquid droplets akin to LLPS-like biomolecular condensates.

We note that HPF1, which regulates PARP1-mediated PARylation on serine residues, had no effect on PARP1 regulation of Polθ end-joining activity in vitro and in cells. However, more than half of PARylated serine residues persist in HPF1 knockout cells, suggesting that PARylation of serines does not rely solely on HPF1[67]. Thus, although we report here that Polθ did not seem to be PARylated on serine(s) in vitro, we cannot exclude the possibility that intracellular Polθ end-joining activity may be affected by serine PARylation.

Despite PARylation-dependent recruitment of Polθ to the vicinity of DNA breaks by PARP1, we observed that PARylation of Polθ abrogated its ability to interact with DNA and chromatin likely due to the strong negative charge of PAR[68]. PARylated PARP1 is also known to dissociate from DNA which supports a general mechanism of PARylation suppressing DNA binding[69]. PARG removal of PAR from Polθ reactivated its interaction with DNA resulting in robust TMEJ activity in vitro and in cells. Thus, DNA/chromatin binding and TMEJ activity of Polθ were restored upon PARG-mediated removal of PAR repressive marks on Polθ. PARG also induced the reversal of PARP1-NAD+ dependent formation of Polθ-DNA biomolecular condensates in vitro.

Since TMEJ and PARP1, and PARG activities stretch through S, G2 and mitosis, it is not unreasonable to speculate that PARG-mediated removal of PAR repressive marks on Polθ regulates TMEJ in various cell cycle stages[4,10–13,70,71]. For example, TMEJ was active in S phase in BRCA2 wild-type cells whereas its activity in BRCA2-null cells was predominant in mitosis[12,13,71].

Taken together, our comprehensive biochemical and cellular data support a two-step mechanism of direct activation of TMEJ by the PARP1-PARG regulatory axis: First, PARP1-dependent PARylation of Polθ facilitates recruitment of inactive PAR-Polθ to the vicinity of DSBs via formation of biomolecular condensates which enables high local concentration of DNA ends, and Polθ (Fig. 5g). Subsequent PARG recruitment to DNA damage foci enables removal of repressive PAR marks on Polθ resulting in its rapid assembly on DSBs to execute TMEJ.

Most recently, Gelot et al.[14] reported that Polo-like kinase 1 (PLK1) phosphorylates Polθ to stimulate TMEJ of the mitotic DSBs. However, TMEJ can also be indirectly stimulated by the PARG-mediated removal of PAR repressive marks on PLK1 kinase. Peng et al.[72] reported that PARP1-mediated PARylation of PLK1 inhibited its enzymatic activity. Since PLK1 phosphorylates Polθ and RHINO to activate TMEJ in BRCA2-null mitotic cells, we can anticipate that PARG-mediated dePARylation of PLK1 would restore TMEJ activity in mitosis[13,14].

The PARP1-PARG-mediated two-step regulation of Polθ and activation of TMEJ described herein may represent a more general phenomenon of DDR regulation. Other DDR pathways are also known to be regulated by PARP1, and similar PARG activation mechanisms may also be at play. For instance, changes in PARylation of BRCA1 was demonstrated as a key factor in the regulation of BRCA1 assembly on chromatin, binding to DNA and HR in prior studies[73]. In concordance, depletion and inhibition of PARG have been shown to inhibit the repair of radiation-induced DSBs[74]. The essential role of PARG in activating TMEJ and the possibility for similar functions for PARG in activating other DDR pathways likely explains why PARG inhibition suppresses the DDR and sensitizes cells to DNA damaging agents[41].

## Methods

### Cell lines

RKO and HEK293T cell lines were from ATCC, MDA-MB-436 cells were used before[75]. U2OS, U2OS-EJ2-GFP (carrying one copy of EJ2-GFP cassette) and U2OS-*HPF1*KO cell lines were described before[60,76]. Cells were cultured in DMEM medium containing 10% FBS and 1% penicillin-streptomycin. FLT3(ITD), JAK2(V617F) and BCR-ABL1 -positive 32Dcl3 cells were described before[77–79]. All the cells were confirmed mycoplasma-free with the MycoAlert Kit according to manufacturer's instructions (Lonza).

### Induction of DNA damage

Experimental cells were exposed to irradiation 10 Gy and incubated for 20 min unless specified. For the Etoposide-induced DNA damage cells were incubated in cell culture medium containing 25 µM etoposide for 30 min before proceeding further.

### Plasmids

pCDH-EF-FHC-POLQ was a gift from Richard Wood (Addgene plasmid #64875: https://www.addgene.org/64875/; RRID: addgene_64875), pCBASceI was a gift from Maria Jasin (Addgene plasmid # 26477: https://www.addgene.org/26477/; RRID: addgene_26477), pLV-mitoDsRed was gift from Pantelis Tsoulfas (Addgene plasmid # 44386: https://www.addgene.org/44386/; RRID: addgene_44386), PARP1 FL plasmid expressing full-length PARP1 (Addgene plasmid #169815, https://www.addgene.org/169815: RRID:Addgene_169815), PARP1 E988Q was created from PARP1 FL using site-directed mutagenesis, 6HIS-GFP-TEV-PARG was a gift from John Tainer, pET28a-SMT_hHPF1 was a gift from John Pascal.

### STING database analysis

Protein-protein interaction database STRING was used to analyze known and predicted Polθ binding proteins.

### PARylation of recombinant Polθ fragments detected by Western blot

PARylation assays of the recombinant Polθ protein fragments were performed as described before[80]. Purified recombinant ΔPolθGS-FLAG (polymerase-helicase domain fusion), Polθ-pol (polymerase domain) or Polθ-hel (helicase domain), 100 ng purified PARP1 (11040-H08B, Sinobiological) was incubated alone or together with purified ΔPolθGS-FLAG (25 ng), Polθ-Pol-FLAG (150 ng), Polθ-Hel (150 ng) for 20 min at 25 °C in 15 µl of activity buffer (25 mM Tris-HCl, pH 7.6, 5 mM MgCl2, 1 mM dithiothreitol, 0.1 µg/µl BSA, 5 ng/µl activated DNA (#80605, BPS Bioscience, USA) and 10 µM NAD+. PARP1-mediated PARylation of purified p53-GST (#14-865, Sigma-Aldrich, USA) was used as a positive control PARylation substrate. Upon completion of incubation, 1:1 2× sample loading buffer was added to the reactions and boiled for 3 min. The reaction products were resolved by SDS-PAGE and analyzed by Western blotting.

### Immunofluorescence (IF) and confocal microscopy

Cells were processed and analyzed for IF as described before[81]. Briefly, cells were fixed with 4% (v/v) paraformaldehyde for 20 min at 4 °C, washed with PBS, permeabilized with 0.5% (v/v) Triton X-100 for 10 min and blocked with PBS containing 3% BSA. Cells were incubated with the same buffer containing primary antibodies for overnight followed at 4 °C by 1 h incubation with secondary antibodies. The incubations were performed in the dark in a humidified chamber. After ×5 washing in PBST for 3 min, slides were mounted in 20 µl mounting media. Cells were visualized and imaged using Leica SP8 Confocal microscope with lasers of 405 nm, 488 nm, 546 nm, and 647 nm.

Briefly, fluorophore emission was collected by an oil immersion at a ×63 objective magnification. The reconstruction of super-resolution image was conducted by using Z-stacking (5 slices per Z-stack with 1 μm) for association of proteins with chromatin or protein-protein co-localization experiments. The composite image of multiple images taken at different focal distances is shown, and images were analyzed using ImageJ 1.53 K software. For quantification, >50 cells were counted for all conditions from two independent experiments. The primary antibodies used for IF were recognizing: γH2AX (Invitrogen, MA5-28007), PARP1 (Santa Cruz, sc-74440 and sc-8007 AF546), PARG (Abcam, ab169639), PAR (Millipore, MABC547), FLAG (Invitrogen, F7425), and HA (Invitrogen, 32-6700). The secondary antibodies were anti-mouse Alexa Fluor 594 (A11062, Life Technologies) and anti-rabbit Alexa Fluor 488 (A11034, Life Technologies). Foci intensities were analyzed in the nuclei ($n = 50$) using ImageJ 1.53 K (as intensity (a.u)).

## Western blotting

Cells were resuspended in IP lysis buffer (Cat. No: 87787, Thermo Scientific, USA) for 30 min on ice. After centrifugation at $15,000 \times g$, for 15 min at 4 °C, the supernatant was collected. Laemmli buffer was added, and the mixture boiled for 3 min. Then, proteins were separated in 4–20% SDS-PAGE (GenScript) and transferred onto Protran BA85 nitrocellulose membrane (Whatman, Germany). Blocking of the membrane was achieved by soaking for 1 h in PBS Tween 0.1% containing 5% non-fat milk. Primary antibodies detecting Polθ (PA5-69577, Sigma), PARP1 (Santa Cruz, sc-74440), PARG (Abcam, ab169639), PAR (Millipore, MABC547), FLAG (Invitrogen, F7425), and actin (MA5-11869, Invitrogen), were added for overnight incubation. Next day, after three times washing with 0.1% TBST, secondary antibodies from LI-COR: IRDye 800CW (926-32210) or IRDye 680CW (926-68073) were added, and following incubation and washing with TBST, blots were scanned using ODYSSEY 3.0.30 software.

## Immunoprecipitation

Immunoprecipitations were performed as described before[82,83]. Briefly, cells transiently expressing Polθ-FLAG-HA were washed once with 1× PBS and centrifuged at $3000 \times g$ for 5 min. Pellet were subjected to lysis using IP lysis buffer. Lysate were precleared using Dynabeads™ Protein A (10001D, Invitrogen) and followed by incubation with anti-HA magnetic beads (88837, Pierce) and anti-PARP1 (Santa Cruz, sc-74440) for 16–18 h at 4 °C. Next day, beads were washed twice with lysis buffer, once with PBS. Purified proteins were eluted with SDS sample buffer by incubating for 15 min at 95 °C. Proteins were resolved on SDS-PAGE and analyzed by Western blotting.

## LC-MS/MS

The samples were analyzed by LC-MS/MS at the Proteomics and Metabolomics facility, Wistar Institute, Philadelphia, PA, USA. LC-MS/MS of tryptic peptides was performed using a nanoACQUITY UPLC (Waters) coupled with a Q Exactive Plus mass spectrometer (Thermo Fisher Scientific). Samples were loaded onto a UPLC Symmetry trap column (180 μm i.d. × 2 cm packed with 5 μm C18 resin; Waters), and peptides were separated by reversed-phase HPLC on a BEH C18 nanocapillary analytical column (75 μm i.d. × 25 cm, 1.7 μm particle size; Waters) using a 90 min gradient formed by solvent A (0.1% formic acid in water) and solvent B (0.1% formic acid in acetonitrile) as follows: 5–30% B over 70 min, 30–80% B over 10 min, and constant 80% B for 10 min. Eluted peptides were analyzed by the mass spectrometer set to repetitively scan m/z from 400 to 2000 in positive ion mode. Full MS spectra were recorded at a resolution of 70,000 in profile mode. Full MS automatic gain control target and maximum injection time were set to 3e6 and 50 ms, respectively. MS2 spectra were recorded at 17,500 resolution and MS2 automatic gain control target and maximum injection time were set to 5e4 and 50 ms, respectively. Data-dependent analysis was performed on the 20 most abundant ions

using an isolation width of 1.5 *m/z* and a minimum threshold of 2e4. Peptide match was set to preferred, and unassigned and singly charged ions were rejected. Dynamic exclusion was set to 25 s. Peptide sequences were identified using MaxQuant 1.6.17.0[84]. MS/MS spectra were searched against a UniProt human protein database (10/2/2020) and a common contaminants database. Precursor mass tolerance was set to 4.5 ppm in the main search, and fragment mass tolerance was set to 20 ppm. Digestion enzyme specificity was set to Trypsin/P with a maximum of 2 missed cleavages. A minimum peptide length of 7 residues was required for identification. Up to 5 modifications per peptide were allowed; acetylation (protein N-terminal), deamidation (Asn), phosphorylation (Ser, Thr, Tyr) and oxidation (Met) were set as variable modifications, and carbamidomethyl (Cys) was set as a fixed modification. Peptide, protein and site false discovery rates (FDR) were both set to 1% based on a target-decoy reverse database. The mass spectrometry proteomics data have been deposited into the MassIVE (https://massive.ucsd.edu/) and ProteomeXchange data repositories with the accession number MSV000092839 and PXD045320, respectively.

## Chromatin extracts

Chromatin bound proteins were isolated using a chromatin extraction kit (Abcam, ab117152) according to the manufacturer's instructions. Briefly, $1 \times 10^7$ cell pellets were washed with two times with PBS, added to 1× working lysis buffer ($1 \times 10^6$ cells/100 μL) containing protease inhibitor and transferred tube on ice for 10 min. For the PARP1 or PARG inhibition experiments, 1 μM PARP1 or 2 μM PARG inhibitors were added for 12 h before harvesting. The samples were vortexed vigorously at maximum speed for 10 sec followed by centrifuged at $5000 \times g$ for 5 min at 4 C. The supernatants were carefully removed, and pellet were resuspended with 500 μL ($1 \times 10^6$ cells/50 μL) of working extraction buffer on ice for 10 min and vortexed occasionally. The resuspended samples were sonicated 2 × 20 sec. The samples were cooled on ice between sonication pulses for 30 sec, and centrifuged at $12,000 \times g$ at 4 °C for 10 min. The supernatants were transferred to a new vial, and chromatin buffer was added at a ratio of 1:1.

## Intracellular TMEJ assays

These assays were performed as described before[20]. Briefly, U2OS cells carrying one copy of E2J-GFP cassette were co-transfected with I-SceI cDNA and dsRED-Mito cDNA (control for transfection efficiency) using lipofectamine 2000 (Invitrogen, Cat No. 11668-500). GFP+ and Red+ cells were analyzed by flow cytometer (Facscanto, BD) followed by Flowjo V.10.10.0 software 72 h after transfection (Supplementary Fig. 8).

## TMEJ assay in nuclear extracts

The assay was performed as described by Dutta et al[66]. with modifications. Briefly, exponentially growing U2OS cells transiently expressing Polθ-FLAG-HA in 60 mm plates (90% confluent) were irradiated with X-rays (10 Gy). After indicated time points of incubation, the irradiated and control cells were harvested for preparation of nuclear extracts. 100 ng I-SceI digested pBabe-hygro-EGFP-MMEJ[85], the repair substrate, mixed with 100 μl nuclear extracts 30 min with gentle shaking at 30 °C followed by incubation at 15 h at 16 °C. 10 μl of mixture were used to XL-10- gold ultracompetent E. coli (Agilent Technologies) following the manufacturer's protocol. The colonies in each agar plate were counted and submitted for PCR + sequence analysis of the product using the PCR primers: 5′-ACGGGGTCATTAGTTCATAGCCCA-3′, and 5′-GGGATTTTG CCGATTTCGGCC-3′; and the sequencing primer: 5′-ATGGTGAGC AAGGGCGAGGAG-3′ (Genewiz Inc.).

## Bone marrow cells from the knockout mice

All mouse experiments were approved by the Institutional Animal Care and Use Committees review board at Temple University. Mice were

housed in a temperature- and light-controlled animal facility under a 12-h light/dark cycle and were provided standard food and water ad libitum. Generation and genotyping of the wild-type, *Polq−/−*, *Parp1−/−* and *Polq−/−;Parp1−/−* mice were described before[20,86]. Briefly, *Polq−/−* mice (purchased from Jaxon Laboratories, JAX #006194) and *Parp1−/−* mice (provided by Roberto Caricchio, Temple University School of Medicine) were cross-bred to generate *Polq−/−; Parp1−/−*, *Polq−/−*, *Parp1−/−* and *wt* animals. Transgenic/knockout mice were identified by PCR of tail snip DNA. DNA isolation and purification from mice tails were performed using the REDExtract-N-Amp Tissue PCR Kit (Sigma-Aldrich). Genotyping for the *Polq* and *Parp1* was performed using 2 × GoTaq polymerase Master Mix (Promega). *Polq*-specific primers used were: wild-type (5′-TGCAGTGTACAGATGTTACTTTT-3′; TGGAGGTA GCATTTCTTCTC-3′) amplified 190-bp fragment, and *Polq* mutant (5′-TCACTAGGTTGGGGTTCTC-3′; 5′-CATCGAAGCTGACTCTAGAG-3′) amplified 300-bp fragment. *Parp1*-specific primers used were: forward, 5′-AGGTGAGATGACAGGAGATC -3′; wild-type reverse, 5′-CCAGCGCA GCTCAGAGAAGCCA-3′; and mutant reverse, 5′-CATGTTCGATGGGA AAGTCCC-3′. The primers amplified a 112-bp fragment if wild-type, a 350-bp fragment if *Parp1* null, and both 112- and 350-bp fragments if heterozygous using amplification conditions for both *Polq* and *Parp1* consisting of 35 cycles at 94 °C for 1 min, 60 °C for 1 min, and 72 °C for 3 min. PCR products were run in a 1.5% agarose gel containing ethidium bromide and visualized using the Gel Doc XR+ Molecular Imager System (Bio-Rad). Lin-mouse bone marrow cells (mBMCs) were obtained as described before[20,86] and cultured in Iscove's MEM containing 10% FBS, 1× antibiotic, 3 ng/ml IL3, IL6 3 ng/ml, and mCSF 5 ng/ml.

## PARP1, PARG, and HPF1 targeting

PARG inhibitor PDD00017273 (SML1781, Sigma), Polθ inhibitor ART558 (HY-141520, MedChemExpress), PARP inhibitor olaparib (S1060, Selleckchem) and PARP1 inhibitor AZD5305 (S9875, Selleckchem) were purchased. Cells were treated with the PARP, PARP1 and PARG inhibitors for 6 h and with Polθ inhibitor for 16 h before transfection. Transient downregulation of PARG and PARP1 was achieved 24 h after transfection with 500 ng of respective shRNA (PARG: TRCN0000050800: TACCAGGGTT ACTGTTTGAGG, PARP1: TRCN0000007930: TTGAGGTAAGAGATTTCTCGG) using Lipofectamine 2000. For HPF1 downregulation, 40 nM HPF1 siRNA (s29883, Life Technologies) were transfected using Lipofectamine 2000.

## Protein expression and purification

Polθ-pol and Polθ-hel were purified as described[25]; PolθΔcen and Fl-Polθ were purified as described[1]. Full-length PARP1 was expressed and purified as described[50], with the following modifications. Briefly, pET28a-based plasmid expressing the N-terminal HIS-tagged full-length PARP1 (1–1016 aa, Addgene plasmid #169815, https://www.addgene.org/169815: RRID:Addgene_169815) was transformed into BL21(DE3) cells (Invitrogen). Freshly grown colonies were inoculated into a starter culture of 30 mL LB supplemented with 50 μg/mL kanamycin and were shaken overnight at 37 °C. Next, the overnight cells were added to 6 L of LB with 50 μg/mL kanamycin and grown at 37 °C until $OD_{600}$ ~ 0.5. Then $ZnCl_2$ was added till 0.1 mM, the shaker temperature was turned to 18 °C, and the cells grew for the next 1 hour followed by addition of IPTG to a final concentration of 0.2 mM. The cells were further shaken overnight and pelleted in a centrifuge at 4 °C (30 min at 3000 × *g*). The pellet (40 g) was next resuspended in 300 mL of lysis buffer containing 25 mM HEPES pH 8.0, 0.5 M NaCl, 20 mM imidazole pH 8.0, 5 mM βME, 0.1% IGEPAL CA-630 supplemented with 2 mM PMSF and 4 tablets of SIGMAFAST EDTA-free protease inhibitor cocktail (Sigma). The cells were sonicated on ice and centrifuged for 60 min at 25,000 × *g*. The cleared lysate was loaded onto a 5 mL HisTrap FF crude column (Cytiva) and washed with lysis buffer and with high salt buffer A (lysis buffer with 1 M NaCl and 1 mM PMSF). The bound protein was eluted with buffer B (lysis buffer with

400 mM imidazole). The fractions containing PARP1 were pooled and dialyzed against 1 L of buffer C (25 mM HEPES pH 8.0, 0.25 M NaCl, 5 mM βME, 0.005% Igepal) overnight at 4 °C. The protein was then loaded onto a 5 mL HiTrap Heparin HP column (Cytiva) and eluted with a NaCl gradient (from 0.25 M to 1 M) in buffer C. Fractions containing PARP1 were pooled, concentrated on a spin concentrator Amicon Ultra with 30,000 MWCO (Sigma), centrifuged 10 min at 20,000 × *g* and loaded onto a size exclusion column Superdex200 Increase 10/300 (GE Healthcare). The desired protein fractions were combined, aliquoted, and frozen at −80 °C. To clone the PARP1 mutant, PARP1 E988Q, the QuikChange XL site-directed mutagenesis kit (Agilent Technologies, cat# 200516-5) was used. PARP1 E988Q was purified using the same protocol as WT PARP1.

pT28a-based plasmid expressing 6HIS-GFP-TEV-PARG catalytic domain (449–962 aa, a gift from John Tainer) was transformed into BL21(DE3) cells. Freshly grown colonies were inoculated into a starter culture of 30 mL LB supplemented with 50 μg/mL kanamycin and were shaken overnight at 37 °C. Next, the overnight cells were added to 4 L of LB with 50 μg/mL kanamycin and grown at 37 °C until $OD_{600}$ ~ 0.5, then the shaker temperature was turned to 18 °C, and the cells were growing for the next 1 h followed by addition of IPTG to a final concentration of 0.2 mM. The cells were further shaken overnight, pelleted in a centrifuge at 4 °C (30 min at 3000 × *g*) and resuspended in 200 mL of lysis buffer containing 50 mM HEPES pH 8.0, 0.5 M NaCl, 10 mM imidazole pH 8.0, 5 mM βME, 0.1% IGEPAL CA-630 supplemented with 2 mM PMSF and 4 tablets of SIGMAFAST EDTA-free protease inhibitor cocktail (Sigma). The cells were sonicated on ice and centrifuged for 60 min at 25,000 × *g*. The cleared lysate was loaded onto a 5 mL HisTrap FF crude column (Cytiva) and washed with lysis buffer with 30 mM imidazole. The bound protein was eluted with lysis buffer with 200 mM imidazole. The fractions containing 6HIS-GFP-TEV-PARG were pooled and dialyzed against 1 L of dialysis buffer (50 mM Tris-HCl pH 7.5, 0.4 M NaCl, 5% glycerol, 5 mM imidazole, 5 mM βME, 0.005% IGEPAL CA-630) with addition of 500 U of TEV Plus protease (Promega Corp) overnight at 4 °C. The protein was then loaded onto a 5 mL HisTrap HP column (Cytiva), and the flow-through fractions containing TEV protease-cleaved untagged PARG (449-962 aa) were collected, pooled, concentrated on a spin concentrator Amicon Ultra with 30,000 MWCO (Sigma), centrifuged 10 min at 20,000 × *g* and further purified on a size exclusion column Superdex200 Increase 10/300 (GE Healthcare). The desired PARG fractions were combined, aliquoted and frozen at −80 °C. Red-fluorescent Polθ-pol was generated using Monolith Protein Labeling (Cat.# NC1561926 Fisher; Nanotemper Technologies Inc; MOL014 RED-fluorescent dye NT-647-NHS).

The human hHPF1 protein was expressed using the plasmid pET28a-SMT_hHPF1 (a gift from John Pascal) and purified as described[52]. The only modification was that the SMT SUMO-like tag was cut using SUMOstar protease (LifeSensors, Cat.# SP4110).

## PARylation of Polθ constructs in vitro detected by SDS gel analysis

To perform PARylation of the recombinant Polθ domains or self-PARylation by the PARP1 in vitro, 0.5 μl of N-terminal His-tagged full-length PARP1, 0.5 μl of pssDNA-1, and 2 mM NAD+ (where indicated) were mixed with the 0.5 μl of Polθ-hel, 0.5 μl of Polθ-pol, or 0.5 μl of BSA (where indicated) in reaction buffer containing 30 mM HEPES pH 8.0, 1.5 mM $MgCl_2$, 1 mM DTT, in total volume of 10 μL followed by 30 min incubation at 30 °C. 6x Laemmly loading buffer was added to stop the reaction, the samples were resolved in 4–15% gradient SDS-PAGE (Bio-Rad) and stained with Coomassie Blue. To perform self-PARylation and PARylation of the Polθ domains in the presence of HPF1, 1 μM of each protein, 1 μM of PAS (PARP1 Activation Substrate) DNA, and 0.5 mM NAD+ were used. Reactions containing DNA, PARP1 and other proteins, where indicated, were incubated for 10 min at

room temperature followed by the addition of NAD+ and further incubated for 30 min at 30 °C. The reactions were stopped with PARP1 inhibitor olaparib at 0.5 mM. Where indicated, reactions were treated with 1 M hydroxylamine ($NH_2OH$) for 60 min at room temperature, then quenched with 0.3% HCl. 6x Laemmly loading buffer was added, and the samples were resolved in 4–15% gradient SDS-PAGE (Bio-Rad) and stained with Coomassie Blue.

### TMEJ reconstitution in vitro

25 nM 5'-$^{32}$P radio-labeled pssDNA-2 (5'-$^{32}$P-labeled RP344: CACTGT-GAGCTTAGGGGTTAGCCCGGG /RP343: CTAAGCTCACAGTG) in reaction buffer (25 mM Tris-HCl pH 7.5, 10 mM $MgCl_2$, 10% glycerol, 0.1 mg/mL BSA, 0,01% IGEPAL CA-630, 1 mM DTT); the PARylation reactions were initiated by the addition of the 10 nM of Polθ variants, and, where indicated, 25 nM of PARP1, 25 nM PARG, 25 nM HPF1 and 100 μM NAD +. The reactions were incubated for 15 min at 37 °C followed by the addition of 50 μM dNTP. Next, TMEJ reactions were incubated for 15 min or as indicated at 37 °C and stopped by the addition of denaturing formamide loading buffer, resolved in denaturing 20% PAAG and analyzed using Typhoon PhosphorImager (GE Amersham). Quantification was done in ImageQuant software. All quantified experiments were performed in triplicates and plotted as mean ± SD using GraphPad Prism 9 software. For XmaI digestion assay, after the initial TMEJ reactions with the 15 nM of Polθ-pol, 25 nM of pssDNA-2, and 50 μM dNTP, 25 Units of XmaI supplemented with 10$^x$ CutSmart digestion buffer (both from New England Biolabs) were added and incubated for further 60 min at 37 °C. The reactions were stopped by the addition of denaturing formamide buffer and resolved as above.

### Polθ-hel ATPase assay

15 nM of Polθ-hel, 15 nM of pssDNA-1, 100 nM of full-length PARP1, and 200 μM NAD+ (where indicated) in reaction buffer (25 mM Tris-HCl pH 7.5, 5 mM $MgCl_2$, 30 mM NaCl, 5% glycerol, 0.1 mg/mL BSA, 1 mM DTT) were incubated for 20 min at 37 °C, followed by the addition of 100 μM ATP and 1–2 μCi of γ-$^{32}$P ATP. After 10 min incubation at 37 °C, the reactions were stopped by the addition of denaturing formamide buffer and resolved in 20% denaturing PAGE. Quantification was done in ImageQuant software; all quantified experiments were performed in triplicates and plotted as mean with ± standard deviation (SD)

### EMSA

40 nM of Polθ-hel, 20 nM of PolθΔcen, 25 nM of PARP1, indicated concentrations of PAS (PARP1 activation DNA substrate), and 100 μM NAD+ were mixed, where indicated, in reaction buffer (25 mM Tris-HCl pH 7.5, 5 mM $MgCl_2$, 30 mM NaCl, 5% glycerol, 0.1 mg/mL BSA, 1 mM DTT) and incubated 20 min at 37 °C followed by the addition of 10 nM of Cy3-labeled ssDNA (RP316-Cy3). After 5–10 min incubation at room temp, glycerol was added till the final 10%, the samples were resolved in non-denaturing 8% PAAG with 0.5× TBE at room temp, and DNA was visualized using Typhoon Phosphor Imager. To test DNA binding inhibition by Polθ-hel or Polθ-pol in the presence of ADP-diphosphate ribose (ADR, Sigma-Aldrich) or poly (ADP-ribose) polymer (PAR, Bio-Techne), 10 nM of Cy3-labeled various DNA templates, 100–150 nM of Polθ-hel or Polθ-pol, and indicated concentrations of ADR or PAR were mixed in the same reaction buffer, incubated for 15 min at room temperature, loaded on the same non-denaturing PAAG and visualized using Typhoon Phosphor Imager. Aurintricarboxylic acid (ATA) was used as a protein-DNA binding inhibition control.

### PARP1-dependent formation of biomolecular condensates

Red-fluorescent Polθ-pol (Cy5 fluorescence) was generated using Monolith Protein Labeling (Cat.# NC1561926 Fisher; Nanotemper Technologies Inc; MOL014 RED-fluorescent dye NT-647-NHS). Pre-mixed Red-Polθ-pol (250 nM), DNA-Cy3 (250 nM), PARP1 (1 μM), PARG (50 nM), PARGi (PDD0001727; 30 μM) and NAD+ (2 mM) were used, as

indicated. After 60–120 min. of incubation reaction aliquots were added into μ-Slide VI 0.4 lbidi Chamber slides and mounted on the stage of a Nikon A1R Confocal Microscope. Each sample was visualized with Plan Fluor ×40 Oil DIC H N2 objective using Nis Elements 5.21 Imaging Software for image capturing. Throughout the experiment, chambers were maintained at 37 °C using a live cell environmental chamber. Cy3 was illuminated using 561 nm laser line and visualized with TRITC (tetramethyl rhodamine) filter sets (0.3% power, 4 gain, 2 frame averaging), while Cy5 dye was illuminated using 640 nm laser line and visualized with Cy5 filter sets (1.5% power, 63 gain, 2 frame averaging). Images were analyzed using ImageJ software. Particles were analyzed based on size (Pixel units–micron$^2$) as well as circularity (0–1). For quantification, two different fields were counted for Cy3, Cy5 and co-localization spots from three independent experiments.

### DNA

The following single-stranded or partially single-stranded DNA (pssDNA) templates were used: RP316-Cy3 (5'-labeled with Cy3), PAS (PARP1 Activation Substrate), pssDNA-1 (LM1/RP663c), pssDNA-2 (5'-$^{32}$P radio-labeled RP344/RP343), pssDNA-3 (5'-Cy3-labeled RP348/RP343), and dsDNA (5'-Cy3-labeled RP348/RP348c). All the oligos including Cy3-labeled were purchased from IDT. 5'-$^{32}$P radiolabeling was done using T4 polynucleotide kinase (New England Biolabs) and γ-$^{32}$P ATP (Perkin Elmer). Labelled pssDNA templates were obtained by annealing labelled vs unlabeled oligos in a ratio of 1:1.5 using 100 °C–25 °C cooling down conditions. The sequences (5'–3') are listed below. RP316-TTTTTTTTTTTTTTTTTTTTTTTTTTTTTTTT; PAS-/5Phos/GCTGGC TTCGTAAGAAGCCAGCTCGCGGTCAGCTTGCTGACCGCG; LM1-GCCT TCATCGCCGAGGAAGGGTGGCTATTGGTGGGCTATACGTTAGTGGCA TCAATCCGC; RP663c-CACCAATAGCCACCCTTCCTCGGCGATGAAG GC; RP343-CTAAGCTCACAGTG;RP344-CACTGTGAGCTTAGGGTTAG CCCGGG; RP348-CACTGTGAGCTTAGGGGTTAGAGCCGG; RP348c- CCG GCTCTAACCCTAAGCTCACAGTG.

**Statistics and reproducibility.** Data are expressed as mean ± standard deviation (SD) from at least three independent experiments unless stated otherwise. Details of the number of events counted are included in each figure. The $p$ value < 0.05 was considered statistically significant.

### Reporting summary

Further information on research design is available in the Nature Portfolio Reporting Summary linked to this article.

## Data availability

The plasmid sequencing data generated in this study was deposited in the figshare. [https://figshare.com/s/8f1254126c624fc79dee]. The mass spectrometry proteomics data have been deposited into the MassIVE (https://massive.ucsd.edu/) and ProteomeXchange data repository with the accession number MSV000092839 [https://massive.ucsd.edu/ProteoSAFe/private-dataset.jsp?task=d7de40a8d4af49a687d6ef86b1c7a279] and PXD045320, respectively. All remaining data supporting the findings of this study are available within the paper and its Supplementary Information. All materials used are described in Supplementary Data 1. Source data are provided in this paper.

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

## Acknowledgements

This work was sponsored by NCI CA186238, CA283396, and CA247707 (to T.S.), CA244179 (to T.S and R.T.P.), R01GM137124 and R35GM152198 (to R.T.P.), and the Leukemia and Lymphoma Society TRP 6628-21 (to T.S.). We thank Dr. Ivan Ahel (University of Oxford, Oxford, UK) for providing U2OS-*HPF1*KO cells.

## Author contributions

T.S. and R.T.P. conceived the idea, supervised the lab project, and wrote the manuscript. U.V., L.M., M.T., T.K., G.C., K.S.-R., J.A., A.-M.K., M.N.-S., D.R., and H.-Y.T. performed the experiments.

## Competing interests

R.T.P. is a co-founder and chief scientific officer of Recombination Therapeutics, LLC. The remaining authors declare no competing interests.
