## [Peer Review File · Nature Communications]

PARG is Essential for Pol θ -Mediated DNA End-Joining by Removing Repressive Poly-ADP Ribose MarksREVIEWER COMMENTS

Reviewer #1 (Remarks to the Author):

While the involvement of PARP1 in alternative end joining repair of double-strand breaks has been known for some time, its exact role has remained unclear. In this manuscript, Vekariya and colleagues use a combination of biochemical, molecular, and cellular techniques to show that poly-ADP ribosylation of DNA polymerase theta assists in its recruitment to chromatin/breaks through the formation of biomolecular condensates. Importantly, they also demonstrate that PARG is required to remove the poly-ADP ribose modifications from pol theta in order to allow it to carry out TMEJ. These findings are important and help to unite a set of disparate observations from many labs into a coherent model describing how temporal regulation of PARP modification of pol theta promotes efficient end joining. The experiments are appropriately controlled and presented in a clear manner. The time course experiments were particularly convincing. There are some outstanding questions that should be addressed to maximize the impact of the work. These, together with a few experimental suggestions, are outlined below.

1. Fig. 1D,E: The immunofluorescence foci show very little overlap in these pictures, suggesting that much of pol theta is not associating with PARP1 at any given time. This could be due to the transient nature of the enzymatic association. I recommend that the authors also calculate and present the percentage of POLQ foci that overlap with PARP foci, to highlight the transient nature of the interaction.
2. Fig. 6: The mechanism behind the formation of the biomolecular condensates in this figure should be further explored by including an inhibitor of PARG. This will rule out the trivial explanation that addition of PARG disrupts phase separation through a non-enzymatic mechanism.
3. Fig. 7: What fraction of the colonies had repair events involving microhomology annealing (MMEJ)? This value (as determined by sequencing) should be provided.
4. Page 19: "The colocalization of POLtheta-FLAG-HA foci increased about 5-fold at 120 min post-irradiation and both PARP1i and PARGi abrogated the colocalization."
How does this observation fit with the model that poly-ADP ribosylation of theta allows it to be recruited to sites of damage? Shouldn't ribosylated pol theta still be able to localize to damaged sites when PARG is inhibited (perhaps even to a greater extent)? This should be explained.
5. If pol theta is coalescing via phase separation together with many DNA ends (Fig. 6C,E,F), wouldn't this greatly increase translocation frequency? Does PARG action somehow prevent this? This point is worth exploring in the discussion.
6. Many of the experiments are performed with either the polymerase domain or the delta-cen protein lacking the central domain. Do the authors have any evidence that the central domain is also poly-ADP ribosylated? If so, given the intrinsically disordered nature of this domain, what effect might this have on phase separation of pol theta, particularly in the biomolecular condensate reactions in Fig. 6?

Minor points:

7. Page 21 – The reference to Figure 5C,E,F should be Figure 6C,E,F.
8. Page 22 – (Figure 7A, insect) should read (Figure 7A, insert)

Reviewer #2 (Remarks to the Author):

Vekariya et al analyze the contributions of the enzyme PARG to the regulation of polymerase theta and theta-mediated end-joining (TMEJ). The results focus on biochemical reconstitution of Polθ activities and an analysis of PARP1/PARG impact on these activities. The biochemical experiments suggest a model where PARP1 modification of Polθ with ADP-ribose prevents Polθ from binding to DNA, and PARG removal of the ADP-ribose modification from Polθ restores the DNA binding and thus activity of Polθ. The experiments are done well and clearly presented. The cell-based experiments make clear observations about the co-localization of the factors PARP1/PARG/Polθ, and the relative timing of the localized concentration of these factors in the nucleus. Many aspects of the cell data are self consistent with a role for PARG in regulating TMEJ. The major issue is that the data does not fully support the model as presented, primarily in the aspect of Polθ modification with ADP-ribose being required for recruitment to damage. This aspect of the model does not really make sense to me, as it seems that Polθ would first need to be located at/near the damage to be modified by PARP1. Instead, could Polθ have disordered regions that associate with the liquid demixed regions produced by PARP1/ADP-ribose? The biochemical experiments clearly show that Polθ is modified by PARP1, and that PARG reverses the modification. But this is not evident in the cell based experiments. The cell experiments do not track the specific modification of Polθ with PAR. The signal shown is likely just PARP1 being automodified. The increase and decrease in PAR signal is attributed to Polθ modification, but this is not really what is being monitored.

Page 22,

>For example, at 20 min post-irradiation we observed an increase in Polθ- PARP1 interaction accompanied by abundant PARylation of Polθ..>

>Subsequent accumulation of Polθ-PARG complexes at 60- and 120-min, however, was accompanied by gradual dePARylation of Polθ..>

The statements above assume that all PAR signal is attached to Polθ, which is not likely to be the case. There needs to be a mechanism for tracking the specific modification of Polθ in cells to make these statements.

Also concerning the model, it was also not clear why the condensates need to be dissolved in order for TMEJ to occur, as the model in Figure 6 shows? Is there a cell-based observation of this?

Overall, the study presents several interesting observations that will advance understanding of the factors regulating TMEJ. But the disconnect between the model and the data needs to be addressed.

Other points:

Controls for Figure 6. If unlabeled DNA is used (no Cy3) with Red-Poltheta, what is the signal in the Cy3 channel?

Page 18, >PARG does not affect the respective TMEJ activity of the three constructs of Poltheta...(Figure 3B,C)>

There only seem to be two Poltheta constructs evaluated in these panels.

For consistency, keep the green and red text for GFP and RFP in Figure 3 panel J

Figure 5C, I am confused why gammaH2AX does not increase over time in this experiment, as it did in the experiment in 5B. HA was being pulled down in panel B, but there was still an increase in FLAG signal, so it is not clear why there is not an increase in gammaH2AX in panel C when it is being pulled down.

Page 20, >This effect correlated with enhanced co-localization of Poltheta-FLAG-HA and PARG but decreased co-localization of Poltheta-FLAG-HA and PARP1 (Figure 5E)>

This statement seems like an error since gammaH2AX is listed as being monitored in Figure 5E.

Page 21, >we hypothesized that PARP1 promotes Polθ recruitment to the vicinity of DNA via PARylation dependent formation of biomolecular liquid condensates, and that PARG would reverse this activity.>

Confusing wording. It sounds like PARG would reverse the recruitment of Poltheta.

Page 22, (Figure 7A, inset)
inset

Page 22, >and this was not associated with increased detection of Polθ on chromatin and γH2AX (Figure 4D,E)>
Figure 5D,E was probably meant here.

Reviewer #3 (Remarks to the Author):

Vekariya and colleagues describe in their manuscript a novel interaction between PolQ and PARP1 and investigate the impact of this modification on PolQ regulation. They identify PARP1-mediated ADP-ribosylation of PolQ as suppression signal for PolQ activity, which is elevated by the action of PARG. The manuscript is well written and logically structures, however, suffers from some issues with the experimental design and lack of controls to fully support the presented conclusion.

The following are some of the major concerns that need correction before a more detailed revision:

1) The authors use 2 different recombinant PARP1 proteins in their study (commercial and mad in-house). They do not indicate which version is used in which experiment and this is problematic as the commercial enzyme carries a C-terminal expression tag that will prevent interaction with HPF1 and thus all conclusions drawn from experiments using this enzyme and HPF1 are necessarily invalid (HPF1 has a clear impact on the PARP1 automodification pattern which cannot be observed). The authors should also perform their experiments either with 32P-NAD⁺ or at least control by Western Blot as CBB stain has low sensitivity and can be mis-leading with regards to the ADP-ribosylation status of a protein.

2) The authors show early on that PARP1 co-IP's with PolQ, but associate the PAR signal in later IP's to PolQ. This is problematic as PARP1 is amongst the most highly modified proteins in the DNA damage response and the observed molecular size range of the PAR signal was in earlier studies associated with PARP1 automodification. This is particularly important as their spatialtemporal observation show near identical kinetics for PARP1 and the PAR signal. Therefore, the authors need to demonstrate that the observed signal can be assigned to PolQ and is not related to PARP1. The problem to judge the validity of the results is compounded by the fact that PARP1 blot for input and IP panels is missing in some panels (e.g. Fig. 1K and L). Furthermore, it would be crucial for the mechanistic understanding of the PolQ ADP-ribosylation to identify the potential modification sites on PolQ and find reliable methods to establish that the observed PAR signal is truly attached to PolQ.

3) Reactions in presence of PARP1 and NAD⁺ will produce a significant amount of PARP1 automodification and the authors should show whether PolQ interacts with the ADP-ribose polymers as other polymerases have been identified as potential PAR binders (e.g. PMID 11016934 & 11016934). One possibility could be addition of protein free PAR to the reactions.

4) The authors state that the observed PolQ inhibition must be due to electrostatic repulsion and thus directly propose a model of liquid demixing at the damage sites. This model is intriguing, however, the authors do not explore other potential regulatory avenues such as modification of residues close to the active site, which could make it e.g. impossible for a substrate to bind due to steric hindrance. This is important as liquid dimixing was observed upon high PAR activation, but whether it occurs under physiological condition is still under investigation and some publication suggest that condition that mimic more physiological conditions produce insufficient polymers for this model to apply or the formation of LLPS regions is dependent additional protein components. In furtherance of their model the authors observe in vitro formation of foci in PARP1/NAD⁺ incubated samples that are interpreted

as biomolecular condensates without appropriate controls and state a higher degree of colocalization of PolQ/DNA with these foci. However, the images shown (Fig. 6A-D) appear to have 100% colocalization between PolQ and DNA, thus do not reflect the quantification presented in Fig. 6E. The authors should validate the nature of the foci formed in vitro and demonstrate that their model (PolQ modification, inability to bind DNA and reactivation by PARG, etc) hold up under much milder conditions (see also 5).

5) Protein amounts in the assays are given in ng instead of concentration, which makes it very difficult to access the NAD+:PARP1 and PARP1:(protein substrate) ratios. Rough estimation indicated that the amount of NAD+ used in these assays is very high and could lead to PARP1 star activity (modification of non-physiological sides) as well as non-physiological PAR chain length that are incomparable to in vivo conditions. The authors would need to show that PolQ can be efficiently modified under milder assay condition and investigate whether short chain modification (or even mono-modification if the authors would utilise the known PARP1 E988Q mutant) have an effect on PolQ activity or if the proposed long chain length with repulsion is required for the drop in PolQ activity.

6) The authors assess co-localisation by confocal microscopy. However, their assay condition increase both size and number of the observed foci and the author do not present any controls or mathematical modelling that supports their findings and rules out that the observed colocalization, which is not very pronounced, is not a result of 'random' overlap due to the amount of signal present in both channels.

7) Fig.2I/J: The authors use apparently a different red fluorophore in the HPF1 KD experiment and omit the data graphs from the supplemental files. The data graphs should be presented and all experiments performed using the fluorophores to allow better compatibility and avoid introduction of experimental errors.

RESPONSES TO THE REVIEWER COMMENTS:

Reviewer #1 (Remarks to the Author):

While the involvement of PARP1 in alternative end joining repair of double-strand breaks has been known for some time, its exact role has remained unclear. In this manuscript, Vekariya and colleagues use a combination of biochemical, molecular, and cellular techniques to show that poly-ADP ribosylation of DNA polymerase theta assists in its recruitment to chromatin/breaks through the formation of biomolecular condensates. Importantly, they also demonstrate that PARG is required to remove the poly-ADP ribose modifications from pol theta in order to allow it to carry out TMEJ. These findings are important and help to unite a set of disparate observations from many labs into a coherent model describing how temporal regulation of PARP modification of pol theta promotes efficient end joining. The experiments are appropriately controlled and presented in a clear manner. The time course experiments were particularly convincing. There are some outstanding questions that should be addressed to maximize the impact of the work. These, together with a few experimental suggestions, are outlined below.

1. Fig. 1D,E: The immunofluorescence foci show very little overlap in these pictures, suggesting that much of pol theta is not associating with PARP1 at any given time. This could be due to the transient nature of the enzymatic association. I recommend that the authors also calculate and present the percentage of POLQ foci that overlap with PARP foci, to highlight the transient nature of the interaction.

RESPONSE: As recommended, the percentages of Pol θ foci that overlap with PARP1 foci have been calculated and are shown in Fig. 1d,e. Approximately 30-50% Pol θ foci co-localized with PARP1 which strongly supports the interaction. Please note that, as expected, the number as well as the percentage of Pol θ +PARP1 foci were significantly higher in homologous recombination deficient MDA-MB-436 cells when compared to 293T cells.

2. Fig. 6: The mechanism behind the formation of the biomolecular condensates in this figure should be further explored by including an inhibitor of PARG. This will rule out the trivial explanation that addition of PARG disrupts phase separation through a non-enzymatic mechanism.

RESPONSE: We thank the reviewer for their insight. We now have added the control (new Fig. 6c,d) showing that the addition of a PARG inhibitor (PARGi) reverses the ability of PARG to inhibit biomolecular condensates formed by PARP1-NAD⁺ liquid liquid phase separation (LLPS).

3. Fig. 7: What fraction of the colonies had repair events involving microhomology annealing (MMEJ)? This value (as determined by sequencing) should be provided.

RESPONSE: All sequenced colonies had MMEJ products. This information is now included on page 15.

4. Page 19: "The colocalization of POLtheta-FLAG-HA foci increased about 5-fold at 120 min post-irradiation and both PARP1i and PARGi abrogated the colocalization."

How does this observation fit with the model that poly-ADP ribosylation of theta allows it to be recruited to sites of damage? Shouldn't ribosylated pol theta still be able to localize to damaged sites when PARG is inhibited (perhaps even to a greater extent)? This should be explained.

RESPONSE: The results in Fig. 3k,l showing that both PARP1i and PARGi abrogate the recruitment of Polθ to γH2AX (immunofluorescent-confocal colocalization and co-IP from the chromatin fractions) triggered the subsequent studies of the spatiotemporal role of PARP1 and PARG in this process. Our comprehensive biochemical and cellular data presented in Figs. 4-7 support a two-step mechanism of recruitment of Polθ to DSBs by the PARP1-PARG regulatory axis resulting in activation of TMEJ. *First*, at 20 min. after irradiation PARP1 dependent PARylation of Polθ facilitates recruitment of inactive PAR-Polθ to the vicinity of DSBs via formation of biomolecular condensates which enables high local concentration of DNA ends and Polθ. At that time, however, intracellular co-localization of Polθ with γH2AX is minimal. *Second*, subsequent recruitment of PARG to DNA damage sites at 120 min. after irradiation enables removal of repressive PAR marks on Polθ resulting in its rapid assembly on DSBs to execute TMEJ. The statement suggesting potential sequential roles of PARP1 and PARG in recruitment of Polθ and activation of TMEJ has been added at the end of the chapter describing the results in Fig. 3 (page 11) and it has been discussed on the bottom of page 17.

5. If pol theta is coalescing via phase separation together with many DNA ends (Fig. 6C,E,F), wouldn't this greatly increase translocation frequency? Does PARG action somehow prevent this? This point is worth exploring in the discussion.

RESPONSE: We now show that the addition of PARG inhibits PARP1-NAD⁺ liquid liquid phase separation of Cy3-DNA and Red-Polθ-pol in new control assays in Fig. 6c,d. The activity of PARG in TMEJ is discussed in the discussion section and results section, and a model is illustrated in Fig. 6h. Our comprehensive data demonstrate that PARG removes repressive PAR marks on Polθ, and dissolves biomolecular condensates. These data demonstrate that PARG reactivates Polθ TMEJ in the vicinity of DNA breaks and allows for Polθ to dissipate following TMEJ.

6. Many of the experiments are performed with either the polymerase domain or the delta-cen protein lacking the central domain. Do the authors have any evidence that the central

domain is also poly-ADP ribosylated? If so, given the intrinsically disordered nature of this domain, what effect might this have on phase separation of pol theta, particularly in the biomolecular condensate reactions in Fig. 6?

RESPONSE: We thank the reviewer for their insight. We have tried for months to purify the central domain of Pol θ as a recombinant protein which is very difficult due to its predicted lack of any secondary structures and is therefore thought to be disordered. We did manage to succeed however with purifying a small quantity to test whether PARP1-NAD⁺ PARylates this domain in vitro in the presence of PARP1-NAD⁺ activating DNA. The results showed that the recombinant Pol θ central domain is not PARylated as shown in new Supplementary Fig. 2c. Interestingly, we did find that the polymerase domain of Pol θ is exclusively PARylated within one or more of its disordered domains see new Supplementary Fig. 2b and description in results section. Although PAR mapping of Pol θ domains and motifs is of interest, we feel that this is beyond the scope of this first paper on PARylation of Pol θ and the major effect of PARG and PARP on Pol θ TMEJ.

Minor points:

7. Page 21 – The reference to Figure 5C,E,F should be Figure 6C,E,F.

RESPONSE: Corrected.

8. Page 22 – (Figure 7A, inset) should read (Figure 7A, insert)

RESPONSE: Corrected.

We hope the reviewer will agree that many newly performed control experiments and careful analysis of our data result in a much more comprehensive study with strong conclusions to support our claims and models.

Reviewer #2 (Remarks to the Author):

Vekariya et al analyze the contributions of the enzyme PARG to the regulation of polymerase theta and theta-mediated end-joining (TMEJ). The results focus on biochemical reconstitution of Pol θ activities and an analysis of PARP1/PARG impact on these activities. The biochemical experiments suggest a model where PARP1 modification of Pol θ with ADP-ribose prevents Pol θ from binding to DNA, and PARG removal of the ADP-ribose modification from Pol θ restores the DNA binding and thus activity of Pol θ . The experiments are done well and clearly presented. The cell-based experiments make clear observations about the co-localization of the factors PARP1/PARG/Pol θ , and the relative timing of the localized concentration of these factors in the nucleus. Many aspects of the cell data are self

consistent with a role for PARG in regulating TMEJ. The major issue is that the data does not fully support the model as presented, primarily in the aspect of Pol θ modification with ADP-ribose being required for recruitment to damage. This aspect of the model does not really make sense to me, as it seems that Pol θ would first need to be located at/near the damage to be modified by PARP1. Instead, could Pol θ have disordered regions that associate with the liquid demixed regions produced by PARP1/ADP-ribose?

RESPONSE: We thank the reviewer for their insight and important question. We have now thoroughly evaluated PARP1-NAD⁺ effects on the formation of biomolecular condensates via induction of liquid liquid phase separation (LLPS). For example, it has been known that PARP1 promotes the recruitment to DNA damage foci in cells, however, the mechanism behind this recruitment has never been investigated. Because PARP1 promotes LLPS via NAD⁺ dependent PARylation, and this process enables recruitment of PARylated proteins to biomolecular condensates, we hypothesized that PARP1-NAD⁺ PARylation of Pol θ would not only form biomolecular condensates (as previously shown in the literature), but also enable recruitment of Pol θ to such condensates. To test this hypothesis directly, we used confocal microscopy to show that PARP1-NAD⁺ promotes LLPS in the presence of Cys-DNA MMEJ model substrate in the new Fig. 6a,d. The addition of PARG however inhibits LLPS due to hydrolysis of the PAR chains (see new Fig. 6c,d). The addition of PARP1-NAD⁺, Cy3-DNA and Red- Pol θ -pol resulted in the formation of liquid droplets containing Red- Pol θ -pol (new Fig. 6b,d). We further found a high degree of colocalization of Red- Pol θ -pol with the Cy3-DNA in the presence of droplets formed by PARP1-NAD⁺ (new Fig. 6f,g). These new data along with our comprehensive biochemical and cellular data support the model presented in Fig. 6h. The model supports our evidence that PARP1-NAD⁺ promotes PARylation of Pol θ and its recruitment to liquid droplets via LLPS at the vicinity of DNA breaks which activates PARP1. since our biochemical data show that PARylated Pol θ cannot bind DNA and perform TMEJ, PARG is needed to hydrolyze the PAR chains which enables rapid activation of Pol θ when it is recruited to the vicinity of DNA breaks via PARP1-NAD⁺ induced LLPS. The high concentration of Pol θ in liquid droplets at the vicinity of DNA breaks allows for rapid and robust TMEJ activity upon PARG hydrolysis of the PAR chains. Upon completion of TMEJ, Pol θ and other DNA repair enzymes such as PARP1 can readily dissipate since the liquid droplets have been dissolved. This elegant process enables a 2-step mechanism whereby LLPS allows for inducing recruitment of PARylated proteins at high concentrations to the vicinity of DNA breaks, then PARG hydrolysis of the PAR chains rapidly activates Pol θ DNA binding and TMEJ activity when it is present at very high concentrations near DNA breaks. finally, Pol θ and other proteins readily dissipate.

The biochemical experiments clearly show that Pol θ is modified by PARP1, and that PARG reverses the modification. But this is not evident in the cell based experiments. The cell experiments do not track the specific modification of Pol θ with PAR. The signal shown is likely just PARP1 being automodified. The increase and decrease in PAR signal is attributed to Pol θ modification, but this is not really what is being monitored.

Page 22,

>For example, at 20 min post-irradiation we observed an increase in Pol θ - PARP1

interaction accompanied by abundant PARylation of Polθ.>

>Subsequent accumulation of Polθ-PARG complexes at 60- and 120-min, however, was accompanied by gradual dePARylation of Polθ.>

The statements above assume that all PAR signal is attached to Polθ, which is not likely to be the case. There needs to be a mechanism for tracking the specific modification of Polθ in cells to make these statements.

RESPONSE: In Fig. 1m we present new data showing that abundant PARylation above and around 300 kDa marker was detected **only** in anti-PARP1 and anti-HA (Polθ-FLAG-HA) immunoprecipitates from cells expressing Polθ-FLAG-HA, consistent with the predicted molecular weight of PARylated Polθ-FLAG-HA. In addition, we developed triple-immunofluorescent staining followed by confocal microscopy to distinguish between PARylated Polθ and Polθ colocalizing with PARylated PARP1. Results presented in Figure 1N demonstrated that yellow foci representing PARylated Polθ were readily detectable and 1.67x more frequent than white foci resulting from Polθ colocalizing with PARylated PARP1. These results are described and discussed on pages 8 and 12. Thus, we postulate that PARylation above and around 300 kDa marker detected in anti-HA immunoprecipitates (Fig. 4c and 5b) and PARylated Polθ detected in cells by double immunofluorescence (Fig. 4b) represented temporal changes of PARylated Polθ. Of note, temporal PARylation of Polθ detected in chromatin extracts was consistent with that visualized by fluorescent immunostaining. Moreover, similar changes of PARylated Polθ was detected in anti-γH2AX immunoprecipitates (Fig. 5c) supporting the temporal modification of PARylated Polθ in the context of its interaction with DSBs.

Also concerning the model, it was also not clear why the condensates need to be dissolved in order for TMEJ to occur, as the model in Figure 6 shows? Is there a cell-based observation of this?

Overall, the study presents several interesting observations that will advance understanding of the factors regulating TMEJ. But the disconnect between the model and the data needs to be addressed.

RESPONSE: Our new data in Fig. 6 along with our comprehensive biochemical and cellular data support the model presented in Fig. 6h. The model supports our evidence that PARP1-NAD⁺ promotes PARylation of Polθ and its recruitment to liquid droplets via LLPS at the vicinity of DNA breaks which activates PARP1. Since our biochemical data show that PARylated Polθ cannot bind DNA and perform TMEJ, PARG is needed to hydrolyze the PAR chains which enables rapid activation of Polθ when it is recruited to the vicinity of DNA breaks via PARP1-NAD⁺ induced LLPS. The high concentration of Polθ in liquid droplets at the vicinity of DNA breaks allows for rapid and robust TMEJ activity upon PARG hydrolysis of the PAR chains which

reactivates Pol θ when at high concentration in the vicinity of DNA breaks in liquid droplets. Upon completion of TMEJ, Pol θ and other DNA repair enzymes such as PARP1 can readily dissipate since the liquid droplets have been dissolved. Our cellular data support this model as follows. PARP1 is known to recruit Pol θ to DNA damage and we confirm this. Our cellular data now show for the first time that PARG is subsequently recruited to DNA damage, resulting in de-PARylation of Pol θ and activation of TMEJ in cells. This elegant process enables a 2 step mechanism whereby PARP1-NAD⁺ induced LLPS allows for rapid recruitment of PARylated proteins at high concentrations to the vicinity of DNA breaks within liquid droplets, then PARG hydrolysis of the PAR chains rapidly activates Pol θ DNA binding and TMEJ activity when it is present at high concentrations near DNA breaks. Finally, Pol θ and other proteins can readily dissipate due to PARG hydrolysis of PAR chains, resulting in dissolving of biomolecular condensates. Thus, altogether the biochemical and cellular data form a comprehensive model of multi-step recruitment and activation of TMEJ by the PARP and PARG protein axis.

Other points:

Controls for Figure 6. If unlabeled DNA is used (no Cy3) with Red-Poltheta, what is the signal in the Cy3 channel?

RESPONSE: We have revised Fig. 6 with a new set of experiments to provide a more comprehensive evaluation of PARP1-NAD⁺ induced liquid condensates and the recruitment of Pol θ to these liquid droplets along with more controls (i.e. PARG inhibitor) as suggested by another reviewer. In all assays in Fig. 6, the Cy3-DNA is used which is a model MMEJ substrate with a 5'conjugated Cy3 fluorophore used to detect LLPS, circularity of particles, particle size, and colocalization with Red-Pol θ -pol (detected with Cy5 excitation and emission).

Page 18, >PARG does not affect the respective TMEJ activity of the three constructs of Poltheta...(Figure 3B,C)>

There only seem to be two Poltheta constructs evaluated in these panels.

RESPONSE: There are three constructs of Pol θ evaluated: Pol θ Δ cen and FL-Pol θ in Fig. 3b and Pol θ -pol in Fig. 3c. Fig. 3b has been modified to avoid the misconception.

For consistency, keep the green and red text for GFP and RFP in Figure 3 panel J

RESPONSE: Corrected.

Figure 5C, I am confused why gammaH2AX does not increase over time in this experiment,

as it did in the experiment in 5B. HA was being pulled down in panel B, but there was still an increase in FLAG signal, so it is not clear why there is not an increase in gammaH2AX in panel C when it is being pulled down.

RESPONSE: Fig. 5c detects Polθ-FLAG-HA, PARP1, PARG and PARylated Polθ-FLAG-HA in anti-γH2AX immunoprecipitates. Therefore, detection of γH2AX might not be quantitative, for example the capacity for γH2AX pulldown could be saturated. Please note that the panel shows temporal kinetic of Polθ-FLAG-HA, PARP1, PARG and PARylated Polθ-FLAG-HA.

Page 20, >This effect correlated with enhanced co-localization of Poltheta-FLAG-HA and PARG but decreased co-localization of Poltheta-FLAG-HA and PARP1 (Figure 5E)>
This statement seems like an error since gammaH2AX is listed as being monitored in Figure 5E.

RESPONSE: We apologize for the error, which has been corrected. Polθ-FLAG-HA was replaced by γH2AX in the text.

Page 21, >we hypothesized that PARP1 promotes Polθ recruitment to the vicinity of DNA via PARylation dependent formation of biomolecular liquid condensates, and that PARG would reverse this activity.>
Confusing wording. It sounds like PARG would reverse the recruitment of Poltheta.

RESPONSE: This confusing phrase has been removed.

Page 22, (Figure 7A, inset)
inset

RESPONSE: Corrected.

Page 22, >and this was not associated with increased detection of Polθ on chromatin and γH2AX (Figure 4D,E)>
Figure 5D,E was probably meant here.

RESPONSE: Corrected.

We hope the reviewer will agree that many newly performed controls and careful analysis of our data result in a much more comprehensive study with strong conclusions to support our claims and models.

Reviewer #3 (Remarks to the Author):

Vekariya and colleagues describe in their manuscript a novel interaction between PolQ and PARP1 and investigate the impact of this modification on PolQ regulation. They identify PARP1-mediated ADP-ribosylation of PolQ as suppression signal for PolQ activity, which is elevated by the action of PARG. The manuscript is well written and logically structures, however, suffers from some issues with the experimental design and lack of controls to fully support the presented conclusion.

The following are some of the major concerns that need correction before a more detailed revision:

1) The authors use 2 different recombinant PARP1 proteins in their study (commercial and mad in-house). They do not indicate which version is used in which experiment and this is problematic as the commercial enzyme carries a C-terminal expression tag that will prevent interaction with HPF1 and thus all conclusions drawn from experiments using this enzyme and HPF1 are necessarily invalid (HPF1 has a clear impact on the PARP1 automodification pattern which cannot be observed).

RESPONSE: Commercial PARP1 was used only the experiments in Fig. 1i and j, which detected PARylated recombinant Pol θ using Western blot and do not investigate the role of HIF1. In-house purified PARP1 was applied in other experiments. This information is included in the Results section on pages 6, and 7, and in Materials and Methods section on pages 18 and 24.

The authors should also perform their experiments either with $^{32}\text{P-NAD}^+$ or at least control by Western Blot as CBB stain has low sensitivity and can be mis-leading with regards to the ADP-ribosylation status of a protein.

RESPONSE: We agree with the Reviewer. In the revised manuscript, we now added all of the requested controls as follows. Since $^{32}\text{P-NAD}^+$ is no longer commercially available, we show that both domains of Pol θ and PARP1 are PARylated via Western blot using an anti-PAR antibody in new Supplementary Fig. 2a.

2) The authors show early on that PARP1 co-IP's with PolQ, but associate the PAR signal in later IP's to PolQ. This is problematic as PARP1 is amongst the most highly modified

proteins in the DNA damage response and the observed molecular size range of the PAR signal was in earlier studies associated with PARP1 automodification. This is particularly important as their spatialtemporal observation show near identical kinetics for PARP1 and the PAR signal. Therefore, the authors need to demonstrate that the observed signal can be assigned to PolQ and is not related to PARP1. The problem to judge the validity of the results is compounded by the fact that PARP1 blot for input and IP panels is missing in some panels (e.g. Fig. 1K and L). Furthermore, it would be crucial for the mechanistic understanding of the PolQ ADP-ribosylation to identify the potential modification sites on PolQ and find reliable methods to establish that the observed PAR signal is truly attached to PolQ.

RESPONSE: Following the Reviewer's request, we present additional evidence that Polθ is PARylated in cells. As expected, irradiation (20 min. earlier) increased detection of PARylated proteins below 250 kDa marker in anti-PARP1 immunoprecipitate (Fig. 1m, lane 3) in concordance with *Masaoka et al., J.Biol.Chem., 287:27648,2012* and *Bian et al., Nature Comm., 10:693,2019*. However, abundant PARylation above and around 300 kDa marker was detected **only** in anti-PARP1 immunoprecipitate from cells expressing Polθ-FLAG-HA (Fig. 1m, lane 5), consistent with the predicted molecular weight of PARylated Polθ-FLAG-HA. Moreover, using 3-color immunostaining and confocal microscopy combined with Z-stacking (5 slices per Z-stack with 1 μm) we were able to distinguish PARylated Polθ (yellow foci) from Polθ colocalizing with PARylated PARP1 (white foci) (Fig. 1n). These results demonstrate that ~60% of yellow foci detected by double staining (anti-Polθ and anti-PAR) in Fig. 4B represented PARylated Polθ whereas ~40% might be generated by Polθ colocalizing with PARylated PARP1. The results are described and discussed on page 8 and 12.

Under the physiological conditions and in response to DNA damage ADP-ribose was detected at least on nine amino acid residues (C, D, E, H, K, R, S, T, Y). Pinpointing Polθ amino acids undergoing PARP1-mediated PARylation is beyond the scope of this work. Nevertheless, we found that PARP1-NAD⁺ exclusively PARylates Polθ polymerase domain within at least one disordered domain as shown in new Supplementary Fig. 2b. For example, we observed that PARP1 is unable to PARylate a mutant version of Polθ-pol lacking five disordered domains (Supplementary Fig. 2b). The mutant version of Polθ lacking 5 disordered domains is active and very soluble and has been previously characterized in our prior studies. Since very specialized mass spec methods are needed to detect ADP ribose on recombinant proteins, we would have to first find a collaborator capable of these specialized methods, then initiate mapping studies which we predict would take another 6-8 months or work. Thus, we believe mapping PARylation sites on Polθ would be more appropriate for a follow-up paper since this field is very competitive. We note that the major advances in our manuscript include the essential role for PARG in promoting TMEJ, and the underlying mechanism in which PARG dePARylation activity is essential for reactivating Polθ activity due to its inability to act on DNA in its PARylated state.

It has been reported that serine PARylation depends on HPF1 (PMID: 28190768). Remarkably, more than a half of PARylated serine residues persist in HPF1 knockout cells suggesting that PARylation of serine does not rely solely on HPF1 (PMID: 34625544).

Although we report here that *in vitro* Pol θ did not seem to be PARylated on serine(s) in the presence of hydroxylamine (Fig. 1h), we cannot exclude the possibility that while Pol θ PARylation, TMEJ, and chromatin/ γ H2AX loading are HPF1-independent (Fig. 1h, 3d,i,j, and 5a,b,d), these functions may still require serine PARylation.

We also added PARP1 blots into the Fig. 1k and l, as requested. In conclusion, we present compelling evidence that PARP1 binds to and PARylates Pol θ .

3) Reactions in presence of PARP1 and NAD⁺ will produce a significant amount of PARP1 automodification and the authors should show whether PolQ interacts with the ADP-ribose polymers as other polymerases have been identified as potential PAR binders (e.g. PMID 11016934 & 11016934). One possibility could be addition of protein free PAR to the reactions.

RESPONSE: We thank the reviewer for their insight. We have now thoroughly examined whether both domains of Pol θ bind recombinant PAR chains *in vitro*. For example, we found that both Pol θ -hel and Pol θ -pol DNA binding activities were suppressed by the addition of recombinant PAR chains *in trans* (see new Supplementary Fig. 3), but not by the addition of ADP ribose (see new Supplementary Fig. 4). This indicates that PAR polyanion chains act as nucleic acid mimics that suppress DNA binding by Pol θ -pol and Pol θ -hel even when added *in trans* at concentrations higher than the enzymes respective DNA substrates.

4) The authors state that the observed PolQ inhibition must be due to electrostatic repulsion and thus directly propose a model of liquid demixing at the damage sites. This model is intriguing, however, the authors do not explore other potential regulatory avenues such as modification of residues close to the active site, which could make it e.g. impossible for a substrate to bind due to steric hindrance. This is important as liquid demixing was observed upon high PAR activation, but whether it occurs under physiological condition is still under investigation and some publications suggest that conditions that mimic more physiological conditions produce insufficient polymers for this model to apply or the formation of LLPS regions is dependent on additional protein components. In furtherance of their model the authors observe *in vitro* formation of foci in PARP1/NAD⁺ incubated samples that are interpreted as biomolecular condensates without appropriate controls and state a higher degree of colocalization of PolQ/DNA with these foci. However, the images shown (Fig. 6A-D) appear to have 100% colocalization between PolQ and DNA, thus do not reflect the quantification presented in Fig. 6E. The authors should validate the nature of the foci

formed *in vitro* and demonstrate that their model (Pol θ modification, inability to bind DNA and reactivation by PARG, etc) hold up under much milder conditions (see also 5).

RESPONSE: We thank the reviewer for their insight. We have repeated the LLPS assays with many more controls, and all of the assays were performed with protein concentrations lower than those used in most published studies investigating LLPS *in vitro* (see new Fig. 6). For example, our new LLPS assays use 250 nM of Pol θ and 1 μ M of PARP1 along with 250 nM of Cy3-DNA. In contrast, most published LLPS papers use significantly higher concentrations of proteins (i.e. 5-50 μ M). We also note that both PARP1 and Pol θ are highly expressed in cancer cells, thus the concentrations used in our LLPS assays are in the proper physiological range for studying their functions. We also note that we titrated down NAD $^{+}$ to very low levels and found strong PARylation of Pol θ *in vitro* under these conditions. For example, only relatively low concentrations of NAD $^{+}$ (<20 μ M) are needed for wild-type PARP1 + NAD $^{+}$ to suppress Pol θ -pol TMEJ via PARP1-NAD $^{+}$ PARylation *in vitro* (see new Supplementary Fig. 5c). Thus, our major finding that PARP1-NAD $^{+}$ PARylation of Pol θ suppresses its TMEJ activity was validated in the presence of low concentrations of NAD $^{+}$ *in vitro*. As suggested by the reviewer, we also fully validated the formation of LLPS liquid droplets by PARP1-NAD $^{+}$ *in vitro*. For example, PARP1 dependent PARylation is known to induce liquid demixing (liquid-liquid phase separation (LLPS)), resulting in sequestration of PARylated and PAR binding proteins into transient and fully reversible spatially confined membrane less compartments, including these at/near DNA damage sites. The circularity or spherical nature of fluorescent particles is widely used as an indicator for the formation of liquid droplets via LLPS. Thus, we used these parameters as a confirmation of liquid droplet formation. As a positive control for liquid droplet formation via LLPS, we first examined whether mixing PARP1 and NAD $^{+}$ with the Cy3-DNA substrate promoted liquid droplets *in vitro* via confocal microscopy. Incubating PARP1 with Cy3-DNA in the absence of NAD $^{+}$ did not result in droplets as expected (see new Fig. 6a, left). For example, only particles with a circularity score <0.9 were observed (new Fig. 6d). The average size of the irregular shaped particles observed under these conditions without NAD $^{+}$ was relatively small (1.78 μ m 2), suggesting they represent PARP1 protein aggregates bound to Cy3-DNA (see new Fig. 6e). We then repeated the experiment but added NAD $^{+}$ which resulted in large droplets and 78.4% of particles exhibited 0.9-1.0 circularity score (see new Fig. 6a, right; Fig. 6d). Particle sizes observed under this condition with NAD $^{+}$ was also substantially larger which is consistent with liquid droplet formation (new Fig. 6e). Next, PARP1 was mixed with Cy3-DNA, NAD $^{+}$ and Red-Pol θ -pol. Again, droplets were clearly observed with 37% of particles exhibiting a circularity score \geq 0.9, and the observation of large particle sizes (new Fig. 6b, left; new Fig. 6d,e). We also observed a high degree of colocalization of Red-Pol θ -pol with Cy3-DNA under these conditions which formed droplets (new Fig. 6b, right; new 6f,g). Since PARylated proteins are known to be recruited to liquid droplets via LLPS, our data indicate that Pol θ -pol is recruited to liquid droplets via PARP1-NAD $^{+}$ PARylation. As a control, we demonstrate that incubation of PARG with PARP1, Cy3-DNA and NAD $^{+}$ prevents the formation of liquid droplets as indicated by no observable particles with a \geq 0.9 circularity score (new Fig. 6c, compare left and center panels; new Fig. 6d). The addition of PARGi, however, reverses this effect, and 96% particles with \geq 0.9 circularity score were observed (new Fig. 6c, right; new Fig. 6d). Hence, these data demonstrate that PARG hydrolysis of PAR chains prevents droplet formation via LLPS. New Fig. 6f,g quantitates the colocalization of Red-Pol θ -pol and Cy3-DNA under conditions with and

without PARP, NAD⁺ and PARG and shows a high degree of colocalization under all conditions, including in conditions with PARP1, NAD⁺ which promotes liquid droplets.

Regarding mechanism of how PARylation of Polθ suppresses its activity, we now performed multiple more controls to probe this based on the reviewer's suggestion. For example, additional Western blots show that the previously characterized PARylation-deficient/MARylation-proficient PARP1(E988Q) mutant MARylates Polθ-pol and Polθ-hel domains as expected (new Supplementary Fig. 5a). We now show that MARylation of Polθ-pol by PARP1(E988Q) does not inhibit its TMEJ activity (new Supplementary Fig. 5b). Hence, PARP1-mediated PARylation, but not MARylation, of Polθ-pol is necessary for suppressing its TMEJ activity. This indicates that the long PAR chains are what suppresses Polθ DNA binding, not single MARylation sites on Polθ. Although the PAR chains on PARP1 via auto-PARylation could potentially inhibit Polθ activity *in trans*, we found that relatively high concentrations of recombinant PAR chains are needed to suppress Polθ activity. For example, we further found that both Polθ-hel and Polθ-pol DNA binding activities are suppressed by the addition of an excess of recombinant PAR chains *in trans* (new Supplementary Fig. 3), but not by the addition of ADP ribose at similar concentrations (new Supplementary Fig. 4). This indicates that PAR polyanion chains act as nucleic acid mimics that suppress DNA binding by Polθ-pol and Polθ-hel when added *in trans* at an excess concentration over their respective DNA substrates. Because an excess of recombinant PAR is needed to suppress Polθ's DNA binding activities, this suggests that the PAR chains bound to Polθ are responsible for inhibiting its DNA binding activities rather than PAR chains *in trans*, such as those attached to PARP1.

5) Protein amounts in the assays are given in ng instead of concentration, which makes it very difficult to access the NAD⁺:PARP1 and PARP1:(protein substrate) ratios. Rough estimation indicated that the amount of NAD⁺ used in these assays is very high and could lead to PARP1 star activity (modification of non-physiological sides) as well as non-physiological PAR chain length that are incomparable to *in vivo* conditions. The authors would need to show that PolQ can be efficiently modified under milder assay condition and investigate whether short chain modification (or even mono-modification if the authors would utilise the known PARP1 E988Q mutant) have an effect on PolQ activity or if the proposed long chain length with repulsion is required for the drop in PolQ activity.

RESPONSE: We again thank the reviewer for their insight and have now performed many more controls as requested. As noted above, we titrated down NAD⁺ to very low levels, and found strong PARylation of Polθ *in vitro* under these conditions. For example, only relatively low concentrations of NAD⁺ (<20 μM) were needed for wild-type PARP1 + NAD⁺ to suppress Polθ-pol TMEJ via PARP1-NAD⁺ PARylation *in vitro* (see new Supplementary Fig. 5c). Thus, our major finding that PARP1-NAD⁺ PARylation of Polθ suppresses its TMEJ activity was validated in the presence of low concentrations of NAD⁺ *in vitro*.

Regarding mechanism of how PARylation of Polθ suppresses its activity, we now performed multiple more controls to probe this based on the reviewer's suggestion. For example, additional Western blots show that the previously characterized PARylation-deficient/MARylation-proficient PARP1(E988Q) mutant MARylates Polθ-pol and Polθ-hel

domains as expected (new Supplementary Fig. 5a). We now show that PARylation of Polθ-pol by PARP1(E988Q) does not inhibit its TMEJ activity (new Supplementary Fig. 5b). Hence, PARP1-mediated PARylation, but not MARylation, of Polθ-pol is necessary for suppressing its TMEJ activity. This indicates that the long PAR chains are what suppresses Polθ DNA binding, not single MARylation sites on Polθ. Although the PAR chains on PARP1 via auto-PARylation could potentially inhibit Polθ activity *in trans*, we found that relatively high concentrations of recombinant PAR chains are needed to suppress Polθ activity. For example, we further found that both Polθ-hel and Polθ-pol DNA binding activities are suppressed by the addition of an excess of recombinant PAR chains *in trans* (new Supplementary Fig. 3), but not by the addition of ADP ribose at similar concentrations (new Supplementary Fig. 4). This indicates that PAR polyanion chains act as nucleic acid mimics that suppress DNA binding by Polθ-pol and Polθ-hel when added *in trans* at an excess concentration over their respective DNA substrates. Because an excess of recombinant PAR is needed to suppress Polθ's DNA binding activities, this suggests that the PAR chains bound to Polθ are responsible for inhibiting its DNA binding activities rather than PAR chains *in trans*, such as those attached to PARP1.

6) The authors assess co-localisation by confocal microscopy. However, their assay condition increase both size and number of the observed foci and the author do not present any controls or mathematical modelling that supports their findings and rules out that the observed colocalization, which is not very pronounced, is not a result of 'random' overlap due to the amount of signal present in both channels.

RESPONSE: Confocal microscopy combined with Z-stacking (5 slices per Z-stack with 1 μm) was performed for protein-protein co-localization experiments. Z-stacking allows to calculate and create the 3D structure of the sample thus minimizing the nonspecific effects. Moreover, analysis of the percentage (not the number shown before) of Polθ foci that co-localize with PARP1 foci (suggested by Reviewer #1) provided more convincing evidence of the interaction; ~35% Polθ foci in 293T cells and ~55% Polθ foci in MDA-MB-436 cells co-localized with PARP1 20 min. after irradiation (Fig. 1d, e; right bottom panels). In addition, please note that as expected the number as well as the percentage of Polθ+PARP1 foci was higher in homologous recombination deficient MDA-MB-436 cells when compared to 293T cells. Altogether, we believe that the colocalization of Polθ foci with PARP1 foci is well documented.

7) Fig.2I/J: The authors use apparently a different red fluorophore in the HPF1 KD experiment and omit the data graphs from the supplemental files. The data graphs should be presented and all experiments performed using the fluorophores to allow better compatibility and avoid introduction of experimental errors.

RESPONSE: We believe that the comment is regarding Fig. 3i/j. We used the same red fluorophore in all experiments, dsRED. dsRED was correctly indicated in the figure legend, but RFP was mentioned by mistake in the panel j; this error has been corrected. Representative supplementary data from HPF1 KD experiment are also included in the Supplementary Fig. 8.

We hope the reviewer will agree that many newly performed controls and careful analysis of our data result in a much more comprehensive study with strong conclusions to support our claims and models.

REVIEWERS' COMMENTS

Reviewer #1 (Remarks to the Author):

Vekariya and colleagues have sufficiently addressed my concerns and provided additional biochemical and cellular data that support their spatiotemporal model of PARP-PARG sequential activity promoting TMEJ. I had the same concern as reviewer 3 that the colocalization of pol theta and PARP might be due to the large increase in foci number/coverage observed with IR, but the new 3-color immunostaining experiments are convincing. The authors have also included a number of additional controls that establish physiological relevance of their findings.

I have only a few minor requested changes for this version.

1. In supplementary figure 5b, it appears to me that MARYlation of Pol theta-pol by PARP1 (E988Q) does partially inhibit TMEJ activity, so the authors should dial back their interpretation in line 256.
2. Line 264: is this referencing the results from Supplementary Fig. 7?
3. Line 290: should refer to Supplementary Fig. 7d.
4. Line 362: should refer to Supplementary Fig. 6.

Reviewer #2 (Remarks to the Author):

The authors have addressed the concerns that I raised during the initial review with more complete explanations in the text.

I detected one error:

Page 13, line 362 "...levels of PARG detection (Supplementary Fig. 7)."
This should be Supp. Fig. 6

And have one comment:

I found that the choice of colors for Cy5 and Cy3 in images made it difficult to interpret the overlays of the two channels. Perhaps one can be set to white to better accent the overlay?

Reviewer #3 (Remarks to the Author):

The improvements Vekariya and colleagues made to the manuscript are well executed and significantly strengthen the manuscript. While not all of my comments could be addressed, I agree with the authors that e.g. identification of PolQ modification sites is beyond the current work and I hope to see this in the future. I believe the manuscript in its current form is of great interest to a wider readership and wish the authors much success in the future work.

RESPONSES TO REVIEWERS' COMMENTS

Reviewer #1 (Remarks to the Author):

Vekariya and colleagues have sufficiently addressed my concerns and provided additional biochemical and cellular data that support their spatiotemporal model of PARP-PARG sequential activity promoting TMEJ. I had the same concern as reviewer 3 that the colocalization of pol theta and PARP might be due to the large increase in foci number/coverage observed with IR, but the new 3-color immunostaining experiments are convincing. The authors have also included a number of additional controls that establish physiological relevance of their findings.

I have only a few minor requested changes for this version.

1. In supplementary figure 5b, it appears to me that MARYlation of Pol theta-pol by PARP1 (E988Q) does partially inhibit TMEJ activity, so the authors should dial back their interpretation in line 256.

RESPONSE: The interpretation in line 251 (revised manuscript) was changed to “We show that MARYlation of Pol θ -pol by PARP1(E988Q) only partially inhibits its TMEJ activity (Supplementary Fig. 5b).” Accordingly, the conclusion in lines 253-254 (revised manuscript) was modified to “Hence, PARP1-mediated PARylation, but not MARYlation, of Pol θ -pol plays a major role in suppressing its TMEJ activity.”

2. Line 264: is this referencing the results from Supplementary Fig. 7?

RESPONSE: The order of Supplementary Figures 6 and 7 was corrected. Line 259 is referencing re-numbered Supplementary Figure 6.

3. Line 290: should refer to Supplementary Fig. 7d.

RESPONSE: The order of Supplementary Figures 6 and 7 was corrected. Line 285 is referencing re-numbered Supplementary Figure 6d.

4. Line 362: should refer to Supplementary Fig. 6.

RESPONSE: The order of Supplementary Figures 6 and 7 was corrected. Line 357 is referencing re-numbered Supplementary Figure 7.

Reviewer #2 (Remarks to the Author):

The authors have addressed the concerns that I raised during the initial review with more complete explanations in the text.

I detected one error:

Page 13, line 362 "...levels of PARG detection (Supplementary Fig. 7)."

This should be Supp. Fig. 6

RESPONSE: The order of Supplementary Figures 6 and 7 was corrected. Line 357 (revised manuscript) is referencing re-numbered Supplementary Figure 7.

And have one comment:

I found that the choice of colors for Cy5 and Cy3 in images made it difficult to interpret the overlays of the two channels. Perhaps one can be set to white to better accent the overlay?

RESPONSE: We thank the reviewer for their insight and fair review. Unfortunately, the microscopy imaging software does not allow for color changes to white. Based on the identical distinctive shapes and identical locations of the particles, and the current distinct colors for Cy3 and Cy5 channels, we believe the particle colocalizations are clear and convincing. We note, the other reviewers did not have issues with the standard colors used by the microscopy imaging software.

Reviewer #3 (Remarks to the Author):

The improvements Vekariya and colleagues made to the manuscript are well executed and significantly strengthen the manuscript. While not all of my comments could be addressed, I agree with the authors that e.g. identification of PolQ modification sites is beyond the current work and I hope to see this in the future. I believe the manuscript in its current form is of great interest to a wider readership and wish the authors much success in the future work.